# Day- and Night-time Formation of Organic Nitrates at a Forested Mountain-site in South West Germany

Nicolas Sobanski[1], Jim Thieser[1], Jan Schuladen[1], Carina Sauvage[1], Wei Song[1], Jonathan Williams[1], Jos Lelieveld[1] and John N. Crowley[1]

[1] Max-Planck-Institut für Chemie, Division of Atmospheric Chemistry, Mainz, Germany.

*Correspondence to*: J. N. Crowley (john.crowley@mpic.de)

**Abstract.** We report in-situ measurement of total peroxy-nitrates ($\Sigma$PNs) and total alkyl nitrates ($\Sigma$ANs) in a forested / urban location at the top of the Kleiner Feldberg mountain in South-West Germany. The data, obtained using Thermal Dissociation

Cavity Ring Down Spectroscopy (TD-CRDS) in August-September 2011 (PARADE campaign) and July 2015 (NOTOMO campaign), represent the first detailed study of $\Sigma$PNs and $\Sigma$ANs over continental Europe. We find that a significant fraction of NO$x$ (up to 75 %) is sequestered as organics nitrates at this site. Further, we also show that the night-time production of alkyl nitrates by reaction of NO$_3$ with biogenic hydrocarbons is comparable to that from day-time, OH-initiated oxidation pathways. The $\Sigma$ANs/ozone ratio obtained during PARADE was used to derive an approximate, average yield of organic

nitrates at noon time from the OH initiated oxidation of VOCs of ~7 % at this site in 2011, which is comparable with that obtained from an analysis of VOCs measured during the campaign. A much lower AN yield, < 2 %, was observed in 2015, which may result from sampling air with different average air mass ages and thus different degrees of breakdown of assumptions used to derive the branching ratio but may also reflect a seasonal change in the VOC mixture at the site.

## 1 Introduction

The gas- and aerosol-phase chemistry of the continental, tropospheric boundary layer is strongly influenced by reactive nitrogen oxides. The primary pollutants NO and NO$_2$ (constituting the NO$x$ family) are mostly emitted by anthropogenic activity involving high temperature combustion or from microbial activity in soils and have a strong impact on tropospheric O$_3$ levels. Knowing the fate of NO$x$ is paramount to prediction of O$_3$ production rates and oxidation capacity on regional and global scales (Lelieveld et al., 2016). Following emission, NO undergoes a series of reaction steps that ultimately lead either

to the formation of short lived trace gases that can act as sinks for NO$x$ (e.g. HNO$_3$) or to the formation of longer lived, reservoir species that can be transported over long disances and act as a source of NO$x$ (e.g. RONO$_2$ and RC(O)OONO$_2$) in locations that are remote from anthropogenic emissions. In the troposphere, a significant amount of NO$x$ can be temporarily sequestered as organic nitrates. HNO$_3$ is produced predominantly by the oxidation of NO$_2$ by OH whereas organic nitrates are produced by the oxidation of both NO and NO$_2$ by organic peroxy radicals during the day (R3 and R4) or in the oxidation

of alkenes by NO₃ during the night (R6) to produce peroxy radicals which subsequently produce stable organonitrates by any radical terminal reaction (R7). Organic peroxy radicals are formed in the oxidation of hydrocarbons by OH (R1 and R2) (Atkinson, 2000; Atkinson and Arey, 2003a, b).

The products of NO$x$ oxidation by organic radicals are peroxy nitrates (RO₂NO₂, abbreviated as PNs, R3) and alkyl nitrates (RONO₂ abbreviated called ANs, R4). Based on thermal lifetimes, PNs may be divided divided in two families, those with a carbonyl group adjacent to the peroxy entity (i.e. RC(O)O₂NO₂) and those without. The former are peroxyacyl nitric anhydrides, generally referered to as peroxyacyl nitrates or PANs, and have lifetimes with respect to thermal dissociation on the order of hours in the midlatitude boundary layer. Non-acyl PNs (RO₂NO₂) such as CH₃O₂NO₂ and HO₂NO₂ thermally decompose on timescales of seconds to minutes at temperatures close to 298 K and are thus only encountered in significant concentrations in cold regions of the atmosphere (Murphy et al., 2004; Browne et al., 2011; Nault et al., 2015). Alkyl nitrates are formed with variable branching ratio in a minor channel (R4) of the reaction between organic peroxy radicals and NO, the greater fraction of this reaction leading to formation of an alkoxy radical and NO₂ (R5) and thus (via NO₂ photolysis) ultimately to O₃ formation. Alkyl nitrates are also formed at night during the NO₃ induced oxidation of unsaturated hydrocarbons, the first reaction step being addition of NO₃ to a double bond followed by formation of a nitrooxy alkylperoxy radical (R6), the fate of which includes reaction with HO₂, other peroxy radicals, NO₃, NO₂ or, if available, NO so that the final products are, hydroxy-, hydroxide- and carbonyl- substituted nitrates as well as dinitrates (Schwantes et al., 2015; IUPAC, 2016). The nitrate yields (per VOC reacted) can be high, especially for biogenic VOCs including the terpenoids, and can exceed 50 % (Ng et al., 2016).

Organic nitrates have highly variable lifetimes (from seconds to days) that are mainly controlled by rates of thermal decomposition and thus temperature for PNs. For ANs, OH oxidation, photolysis and deposition or scavenging by aerosol particles all play a role (Roberts, 1990; Browne et al., 2013; Rollins et al., 2013).

$$OH + RH\ (+ O_2) \quad \rightarrow \quad RO_2 + H_2O \quad\quad\quad (R1)$$
$$R=R' + OH\ (+ O_2) \quad \rightarrow \quad R(OH)R'O_2 \quad\quad\quad (R2)$$
$$RO_2 + NO_2 + M \quad \rightarrow \quad RO_2NO_2 + M \quad\quad\quad (R3)$$
$$RO_2 + NO + M \quad \rightarrow \quad RONO_2 + M \quad\quad\quad (R4)$$
$$RO_2 + NO \quad \rightarrow \quad RO + NO_2 \quad\quad\quad (R5)$$
$$R=R' + NO_3\ (+ O_2) \quad \rightarrow \quad R(ONO_2)R'O_2 \quad\quad\quad (R6)$$
$$R(ONO_2)R'O_2\ (+RO x, NO x) \quad \rightarrow \quad ANs \quad\quad\quad (R7)$$

The first field measurements of organic nitrates were made using low time resolution methods (gas chromatography) (Roberts et al., 1989; Blanchard et al., 1993) and focused mostly on PAN (peroxy acetyl nitrate, CH₃C(O)O₂NO₂), PPN (peroxy propyl nitrate, C₂H₅C(O)O₂NO₂) and MPAN (methacryloyl peroxynitrate, CH₃C(CH₂)C(O)O₂NO₂) (Williams et al.,

1997; Williams et al., 2000) which are the most common peroxy nitrates in the continental boundary layer, and individual mono- and polyfunctional alkyl nitrates from alkane or alkene precursors (Atlas, 1988; Flocke et al., 1991). More recently, mass spectrometry based methods have been developed to measure a wider range of organic nitrates at high time resolution (Beaver et al., 2012; Lee et al., 2016b). Early attempts to compare total reactive nitrogen $NO_y$ (where $NO_y = NO_x +$
$RO_2NO_2 + RONO_2 + HNO_3 + HONO + +$) with the sum of individually measured species (Fahey et al., 1986; Buhr et al., 1990; Ridley et al., 1990) revealed that a substantial fraction of $NO_y$ was missing. This phenomenon was addresssed by the first measurements of total peroxynitrates ($\Sigma$PNs) and total alkyl nitrates ($\Sigma$ANs) by thermal dissociation coupled with laser induced fluorescence detection of $NO_2$ (Day et al., 2003). Those and subsequent measurements (Wooldridge et al., 2010; Perring et al., 2013) indicated that, depending on environment, the interaction between $NO_x$ and VOCs leads to a wide
variety of compounds with different levels of structural functionality and atmospheric lifetime and which can have a variable and significant influence on the lifetime of $NO_x$ and particle composition (Browne et al., 2013; Ayres et al., 2015).

We present here an analysis of organic nitrates and $NO_2$ measured using Thermal Dissociation Cavity Ring-Down Spectroscopy (TD-CRDS) during two field campaigns that took place at a forested, semi-rural mountain site in South-Western Germany. As far as we are aware, this work constitutes the first measurement and analysis of $\Sigma$PNs and $\Sigma$ANs over
continental Europe. We show that the daily variations of $\Sigma$ANs and $\Sigma$PNs are controlled by photo-chemical oxidation of VOCs, night-time production by $NO_3$ induced oxidation of biogenic VOCs and local meteorology. We report an estimation of the effective branching ratios, $\alpha$(OH) of OH-induced AN formation using $O_3$ measurements and compare this with an estimation based on VOCs at the site and known, individual branching ratios. The results from the two campaigns are compared and differences discussed in terms of annual and seasonal changes in meteorology and VOC (biogenic / and
anthropogenic).

## 2 Campaign site and meteorology

The August-September 2011 PARADE campaign (PArticles and RAdicals: Diel observations of the impact of urban and biogenic Emissions) and the July-2015 NOTOMO campaign (NOcturnal chemistry at the Taunus Observatory: insights into Mechanisms of Oxidation) both took place at the Taunus Observatory (50.22 N, 8.45 E) on top of the "Kleiner Feldberg"
mountain, 850 m above sea-level and 500 m above nearby urban centres in the states of Hessen and Rheinland-Pfalz in South-Western Germany. This site has been described extensively in different publications (Crowley et al., 2010; Sobanski et al., 2016b) and only a short description is given here. A few km to the NNE and SE of the station are two mountains of similar height ("Großer Feldberg" 878 m and "Altkoenig" 798 m ASL). The nearby (10s of km) environment in the complete northern sector is a partially forested, rural region. The SW-SE sector is a more densely populated, industrialized region
which includes the Frankfurt-Mainz-Wiesbaden urban agglomeration. Frankfurt is situated $\approx$ 20 km to the SE and Mainz and Wiesbaden $\approx$ 20 – 30 km to the SW. A detailed land-use analysis of the surrounding area was given by Sobanski et al. (2016b).

Reactive trace-gas measurements at the site are strongly influenced by the horizontal advection of different types of air masses, both on a local scale (forest/rural vs. urban) and on regional scales (continental vs. marine). On some nights during PARADE, the instruments sampled air from a low lying residual layer which resulted in very high $NO_3$ steady-state lifetimes ($\approx 1h$). Otherwise the $NO_3$ lifetimes were generally less than 10 mins (Sobanski et al., 2016b).

During both campaigns, the meteorological conditions were very variable and associated with different air mass origins. The PARADE campaign can be divided in three periods (Phillips et al., 2012). The first period from the 15[th] to the 26[th] of August was influenced by air masses of continental origin and was associated with high temperature, and low humidity. A cold front arriving from the West resulted in two consecutive days of rain/fog conditions and a large decrease in temperature and ozone. The period 26[th] to 5[th] of September was influenced by advection from the Atlantic / UK region and during this period

the temperature increased progressively together with ozone. A second cold front on the 5[th] of September again resulted in a fast temperature decrease followed by a period of low photochemical activity. The NOTOMO campaign was characterised by frequent fluctuation between cold/wet and warm/dry periods. Back trajectory calculations (48 hrs) showed that the warm/dry periods were generally associated with air masses of continental origin, the cold/wet periods with air masses coming from the West with Atlantic influence.

**3 Instrumentation**

The instruments deployed during both field campaigns have been described in Schuster et al. (2009) and Thieser et al. (2016) for PARADE and in Sobanski et al. (2016a) for NOTOMO. During the PARADE campaign, all instruments used collocated inlets of PFA piping. During the NOTOMO campaign, the instruments described here sampled from a common, high volume-flow inlet. Temperature, wind speed and wind direction data during NOTOMO were measured by the permanent

instrumentation of the Hessian Agency for Nature Conservation, Environment and Geology (HLNUG) at this site.

**3.1 $NO_2$, $NO_3$, ΣPNs and ΣANs**

$NO_2$ and total gas-phase organic nitrates were measured during both campaigns by TD-CRDS. Membrane filters were used to prevent aerosol from entering the CRD inlets, which would lead to severe reductions in the detection limit, degradation of the cavity mirrors and also to the detection of particulate nitrate (both organic and inorganic) in the TD channels.

**PARADE**: During PARADE, a two cavity TD-CRDS instrument was deployed (Thieser et al., 2016). The instrument was located in a container and sampled air from a 5 m ½" PFA tube acting as a bypass flow through which ambient air was drawn at $\approx 40$ dm$^3$ (STP) min$^{-1}$, (hereafter SLM). The Teflon coated (DuPont, FEP, TE 9568) cavities were operated at 405 and 409 nm, and both were maintained at 35 °C to improve thermal stability. One cavity sampled air from the bypass flow at

ambient temperature to measure $NO_2$ mixing ratios, the second channel sampled alternately through two heated inlets, one held at 200 °C and the other at 450 °C to thermally dissociate PNs and ANs respectively into $NO_2$. When sampling through the 200 °C inlet, this channel measures the sum of ambient $NO_2 + $ΣPNs. Sampling via the 450 °C inlet results in detection of

$NO_2 + \Sigma PNs + \Sigma ANs$. Mixing ratios of $\Sigma PNs$ and $\Sigma ANs$ were obtained by the difference in $NO_2$ measured in the two cavities and applying corrections to account for conversion (to $NO_2$) of $ClNO_2$ (measured by CIMS, Phillips et al. (2012)) and $N_2O_5$ (TD-CRDS, Sobanski et al. (2016b)) in the hot inlets and also for biases related to reactions involving peroxy radicals, NO, $NO_2$, and $O_3$ as outlined in detail in Thieser et al. (2016). The average correction factor for the ANs was 1.2, with maxumim

values of 2. The uncertainty associated with the correction procedure is estimated as ~ 30 % (Thieser et al., 2016).

During PARADE, $NO_3$ was measured by the 662 nm, two channel TD-CRDS described in Schuster et al. (2009). One cavity sampled air at ambient temperature and measured $NO_3$, the second one measured the sum of $NO_3$ and $N_2O_5$ following the thermal dissociation of $N_2O_5$ to $NO_3$ and $NO_2$ at 100 °C. This instrument was located on the roof top and sampled air from the centre of a bypass flow (50 SLM) through a 1 m long, ½-inch (12.7 mm) diameter PFA pipe. The detection limit for $NO_3$

and $NO_2$ were 2 and 30 pptv and the uncertainties 15 and 6 % respectively. As described in Thieser et al. (2016), the uncertainty associated with the $NO_2$ measurement is 6 % + 20pptv × RH/100 (where RH is the relative humidity in %). The detection limit and uncertainties for the organic nitrate meaurements depends on NO$x$ levels and the reader is referred to Thieser et al. (2016) for more details.

**NOTOMO:** The 5 channel, TD-CRDS deployed during the NOTOMO campaign was recently decribed in detail by

Sobanski et al. (2016a). This instrument has five identical cavities sampling from separate inlet lines. Two cavities operate at 662 nm for the detection of $NO_3$, the other three at 405 nm to detect $NO_2$. One 662 nm cavity samples 8 SLM from an inlet at ambient temperature to measure $NO_3$, the other draws 7 SLM through an inlet at 110 °C to measure the sum of $NO_3 + N_2O_5$ following thermal dissociation of $N_2O_5$ to $NO_3$. Of the three 405 nm cavities, one draws 2.5 SLM via an inlet at ambient temperature to measure $NO_2$, the other two each sample 2.5 SLM from inlets heated to 175 °C and 375 °C to

measure $NO_2 + \Sigma PNs$ and $NO_2 + \Sigma PNs + \Sigma ANs$, respectively. The ¼ inch (6.35 mm) inlet line for the 662 nm cavities was attached ia a T-piece to a 60 cm long ½ inch PFA pipe sampling air at 100 L min$^{-1}$ from the center of a large diameter (15 cm), high-flow inlet (10 m$^3$ min$^{-1}$), with its opening located 8 m above the ground and 3m above the top of the container. The 405 nm channels sampled air via a 1 m long, ¼ inch PFA tube protruding into the center of the high-flow inlet. Correction for $ClNO_2$ and $N_2O_5$ conversion to $NO_2$ were carried out as described for the PARADE campaign, the removal of the biases

related to reactions by peroxy radicals, $O_3$ and NO$x$ were carried out as described in Sobanski et al. (2016a). The average correction factor for the $\Sigma ANs$ was 1.1, with maximum values of 2. The uncertainty associated with the correction procedure is estimated as ~ 30 % (Thieser et al., 2016). The detection limits were 1.5 and and 60 pptv for $NO_3$ and $NO_2$, respectively, with uncertainties of 25 % for $NO_3$ and 6.5 % for $NO_2$ (Sobanski et al., 2016a).

### 3.2 NO during PARADE / NOTOMO

During PARADE, NO measurements were made with a modified commercial chemiluminescence detector (CLD 790 SR), the operation of which is described by Li et al. (2015). The detection limit for this instrument is 4 pptv in 2 s with a total uncertainty of 4 %. This instrument did not participate in NOTOMO and daytime NO mixing ratios were calculated from measurements of $NO_2$, $O_3$ and $J(NO_2)$ assuming photo stationary state:

$$[NO]_{calc} = J(NO_2) [NO_2] / k_{(NO+O3)}[O_3] \tag{1}$$

where $J(NO_2)$ is the photolysis frequency of $NO_2$ (measured using a METCON spectral radiometer) and $k_{(NO+O3)}$ is the
temperature dependent rate constant for reaction of NO with $O_3$. This expression ignores the oxidation of NO to $NO_2$ via e.g.
reactions of peroxy radicals and thus overestimates NO. However, this method of estimating [NO] resulted in satisfactory
agreement (within $\approx 20$ %) with measurements from the HLNUG for periods when NO was above the detection limit (>
1ppb) of their instrument. Night-time concentrations of NO during NOTOMO were assumed to be zero, consistent with
measurements on many nights during previous campaigns at this site (Crowley et al., 2010; Sobanski et al., 2016b).

**3.3 VOCs measurements during PARADE**

The VOC measurements have been described already by Sobanski et al. (2016b). Briefly, VOCs were measured using two
gas-chromatographic instruments (1 data point per hour) with a mass spectrometer (GC-MS) and a flame ionisation detector
(GC-FID). The GC-MS (biogenic and aromatic hydrocarbons) had a detection limit of around 1 pptv with an uncertainty of
10–15 %. The GC-FID (non-methane hydrocarbons) had detection limits between 1 and 5 pptv, exceptions being ethane,
ethene, propene, benzene and toluene with values of 8, 16, 9, 14 and 48 pptv respectively and a total uncertainty of 10 % (15
% for 1-pentane). The GC-measurements for butadiene and pentene were unrealistically high, probably a result of poor-
separation of trace-gases with similar retention times, and are not reported. The VOCs measured during PARADE are listed
in Table 1 along with their rate constants, $k_{OH+VOC}$, for reaction with OH) and also the associated alkyl nitrate yields, $\alpha(AN)$.
We also list calculated production rates of ANs ($P(ANs)$), and $O_3$ ($P(O_3)$) derived from midday OH levels during PARADE.

**4 Results and discussion**

**4. 1 NO*x* and organic nitrates at the Taunus Observatory**

The temperature, humidity, wind direction, $O_3$, NO, $NO_2$, $\Sigma$PNs and $\Sigma$ANs for the PARADE and NOTOMO campaigns are
shown in Fig. 1 and Fig. 2, respectively. The reactive nitrogen species are plotted at 10 min resolution, and Table 2
summarizes selected minimum, maximum and mean values for the two datasets. The data provided by the HLNUG are given
at 30 min intervals.

During PARADE and NOTOMO, $NO_2$ varied between $\approx 1$ to 15 ppbv with the highest mixing ratios (during PARADE)
associated with air mass originating from the south and from the south - west, corresponding approximately to the urbanized
Frankfurt and Wiesbaden/Mainz sectors. The $\Sigma$PNs mixing ratio varied from below the detection limit to $\approx 3$ ppbv during
NOTOMO. Low values approaching or below the detection limit were measured during episodes of persistent fog and
rainfall. The campaign means for $\Sigma$PNs were 505 pptv for PARADE and 677 pptv for NOTOMO, respectively. $\Sigma$ANs

mixing ratios varied from below the detection limit to 1.2 ppbv (PARADE maximum) with campaign mean values of 297 pptv for PARADE and 116 pptv for NOTOMO, repectively. Figures 1 and 2 show that [ΣPNs] and [ΣANs] covary at the Taunus Observatory and also show a correlation with $O_3$ both in terms of their diel profile (see below) and day-to-day variability.

Measurements of individual peroxyacyl nitrates are numerous, especially of PAN which usually represents ∼ 80% of all peroxy-nitrates in the lower troposphere (Roberts, 1990; Roberts et al., 1998). This has been shown to also be the case at the Taunus Observatory (Thieser et al. 2016). Our measurements of [ΣPNs] are in the range of [PAN] or the sum of individual PANs mixing ratios observed in different urban/suburban locations (Roberts, 1990; Roberts et al., 2007; LaFranchi et al., 2009). Measurements of individual alkyl nitrates are more sparse. Early estimates for the total alkyl nitrate burden at rural

locations influenced by urban emissions up to a few hundred pptv were derived by summing the mixing ratio of individually measured alkyl nitrates (C1 up to C8) by gas chromatography technics (Flocke et al., 1991; Flocke et al., 1998; Russo et al., 2010; Worton et al., 2010). Recent developments in chemical ionisation mass spectrometry have enabled measurement of more complex, multifunctionnal nitrates of biogenic origin and revealed the occassional presence of ppbv levels of the sum of measured alkyl nitrates (Beaver et al., 2012). Our measurements of ΣPNs and ΣANs can be more directly compared to

those of the University of Califiornia at Berkeley, who have developed and applied the technique of TD-LIF over the last 15 years (Day et al., 2002; Wooldridge et al., 2010; Perring et al., 2013). The mixing ratios of ΣPNs and ΣANs at the Taunus Observatory are comparable to summertime measurements of ΣPNs and ΣANs (0 – 2 ppbv and 0 – 1 ppbv, respectively) at forested sites in California (Day et al., 2003; Murphy et al., 2006; Day et al., 2008).

## 4.2 Sequestering of NOx as organic nitrates

In this section, we examine how much NOx is sequestered as organic nitrates at this site. As $HNO_3$ was not measured during either campaign, we cannot examine the relative efficiency of NOx conversion to organic nitrates compared to inorganic $HNO_3$, though we expect the former to dominate in air masses in the continental boundary layer that have significant anthropogenic or biogenic emissions of hydrocarbons (Day et al., 2008).

Figure 3 shows the fraction, f(NOx), of NOx sequestered as organic nitrates (f(NOx) = ([ΣPNs] + [ΣANs]) / ([ΣPNs] +

[ΣANs] + [NOx])), plotted versus [NOx] and colour coded for temperature. It clearly shows that f(NOx) is higher at low NOx, and could be as high as 0.75 in air masses containing less than 2 ppbv of NOx. The lowest values of f(NOx) measured (< 0.08) were associated with levels of NOx in excess of 10 ppbv. At this location, high values of $NO_2$ are associated with freshly emitted anthopogenic pollution originating from the nearby urban centres. The high values of f(NOx) during periods of low NOx is the result of efficient conversion of NOx to longer lived organic nitrates in photochemically aged air masses at

this site. Furthermore, the temperature dependence indicates that organic nitrate formation is not limited by NOx. For a given level of NOx, f(NOx) is larger when temperatures are higher, reflecting stronger biogenic emissions, and more intense photochemical activity and thus conversion rates of NOx to organic nitrates (Olszyna et al., 1994; Day et al., 2008).

However, note that low temperatures also increase the rate of transfer of soluble organic nitrates to the aerosol phase, which acts on $f(NOx)$ in the same direction.

### 4.3 Diel profiles: Photochemical and meteorological influences

Figure 4 shows the mean diel profiles of $J(O^1D)$, $[O_3]$, temperature, humidity, $[NO_2]$, $[\Sigma PNs]$ and $[\Sigma ANs]$ for the PARADE (dashed lines) and NOTOMO (solid lines) campaigns. A number of factors including highly variable (temporal and spatial) local emisions, irradiance, wind-direction and the complex topography at the site all impose their influence on the diel profiles measured for the organic nitrates.

The campaign averaged, daily maxima in global radiation, temperature and $O_3$ mixing ratio were higher during NOTOMO, indicating on average warmer, sunnier and drier conditions in Jul. 2015. Lower average levels of $NO_2$ were measured during NOTOMO. The mean PARADE $NO_2$ profile shows two maxima during the day (at ~ 10:00 UTC and ~ 19:00 UTC) which correspond approximately to local rush-hour traffic increases during the working week. These $NO_2$ maxima are less clearly defined but still apparent in the averaged diel profiles in NOTOMO. A close inspection of the NOTOMO data revealed that the days can be divided into two types, 7 which display the rush-hour peaks in $NO_2$ and 23 which do not.

As illustrated in Fig. 5, rush-hour influenced days were associated with higher temperaures and levels of $O_3$ and a local wind direction that has a large, daytime component from the South, whereas the other days were cooler, had less $O_3$ and the local wind had a dominant westerly component. Air masses arriving from the southerly sectors are influenced by rush-hour traffic from the nearby urban centres, whereas those arriving from the west are cleaner, with an Atlantic influence and more distant emissions of $NOx$. Up-slope winds, resulting from enhanced rates of heating of the mountainside during the warmer periods can also play a role in enhancing rates of transport of $NOx$ and photochemically produced trace gases to the site compared to the cooler, cloudier days under the influence of westerly winds. The mean $\Sigma PNs$ profiles during both campaigns (Fig. 4) indicate an increase in mixing ratio starting at about sunrise with a broad daily maxima between $\approx$ 12:00 and 14:00 for NOTOMO. The mean daily maximum for NOTOMO was about 1.2 ppbv, a factor of two or more than for PARADE. In contrast, during PARADE the mean daily maximum of $\Sigma ANs$ ($\approx$ 0.6 ppbv) was about a factor of three larger than during NOTOMO. The ratio of the mean daily maximum of ANs to PNs, ($[ANs]_{max}$ / $[PNs]_{max}$) was thus very different for the two campaigns, with values of close to one for PARADE and $\approx$ 0.2 for NOTOMO.

A number of factors influence the relative concentrations of PNs and ANs. In general, higher temperatures are the result of higher levels of insolation and are thus usually related to higher $O_3$ concentrations and rates of photochemical processing of VOCs. This should lead to higher concentrations of both PNs and ANs. Higher levels of insolation will lead to higher NO to $NO_2$ ratios (noon-time $NO_2$-to-NO ratios were 4.0 (PARADE) and 3.8 (NOTOMO) and, given sufficient NO, elevated temperaures will reduce the lifetimes of PNs. Altogether, higher temperatures and more insolation favour AN production over PN production. During the two campaigns. This is essentially the opposite to what we observe and we conclude that other factors, including the mechanism of organic nitrate production from oxidation of different VOC types and rates of loss of the organic nitrates play a major role in contolling the relative abundance of ANs and PNs at this site (see below).

## 4.4 Daytime and night-time production of alkyl nitrates

During PARADE, the diel profiles of [ΣANs] show a maximum around 12:00 UTC, similar to the maximum in global radiation which drives primary OH formation, VOCs oxidation and peroxy radical production rates. However, as indicated in Sect. 1, ANs can also be formed by the reaction of $NO_3$ radicals with biogenically emitted VOCs, which can impact on their diel profile. Figure 6 shows median profiles of [ΣANs] obtained by filtering out periods with fog and rain at the site in which $NO_3$ would have been absent due to the rapid, heterogeneous scavenging of $N_2O_5$, with which it is in thermal equililibrium. We also plot the mean profiles of $NO_3$ during these nights (representing 16 "dry" nights for PARADE and 13 "dry" nights for NOTOMO). The most notable change compared to Fig. 4 is the increase in [ΣANs] during the night. For the NOTOMO campaign, the night-time [ΣANs] represent ~60 % of the day-time value, for PARADE this is ~35%. As described in Thieser et al. (2016) and Sobanski et al. (2016a), the potential artefact caused by thermal decomposition of either $ClNO_2$ or $N_2O_5$ (present during some nights in amounts up to several hundred pptv) in the hot inlets of the TD-CRDs was accounted for by the simultaneous measurement of both of these trace gases and therefore does not contribute to the night-time signal we ascribe to ANs.

We now explore potential meteorological and chemical contributions to the night-time increases in [ΣANs]. Sobanski et al. (2016b) report occassionally extended $NO_3$ lifetimes (> 1000 s) at this site that result from sampling from a low-lying residual layer. Compared to the lowest levels, the residual layer is likely to contain higher levels of photochemically generated trace gases (e.g. ANs) which would otherwise be lost by deposition. During PARADE and NOTOMO, the majority of nights were however characterised by $NO_3$ lifetimes of the order of minutes and less, which indicate that $NO_3$ is removed by reaction with VOCs, presumably mainly reactive terpenoids with double bonds. Those nights (altogether 4) with long $NO_3$ lifetimes during PARADE were excluded for calculating the [$NO_3$ ] and [ΣANs] profiles in Fig. 6.

The reaction between $NO_3$ and unsaturated VOCs is known to produce alkyl nitrates with a higher yield than the day-time pathway through OH induced oxidation of VOCs and is a plausible explanation of the night-time maxima in ΣANs shown in Fig. 6. In order to assess this, we calculated the night-time and day-time and production of alkyl nitrates during PARADE as described below.

### 4.4.1 Night-time generation of ANs in PARADE via $NO_3$ reactions

To estimate the night time-production of ANs, we consider the reaction between $NO_2$ and $O_3$ to be the only $NO_3$ precursor. Of the measured VOCs, isoprene, α-pinene, myrcene and limonene account for > 95% of the $NO_3$ reactivity and have substantial yields of ANs. The mean night-time mixing ratios for these four compounds during PARADE (excluding data where RH > 92 %) are listed in Table 3 along with the corresponding $NO_3$-reaction rate coefficients ($k$) and the alkyl nitrate yields ($\alpha$) as reported by the IUPAC panel (IUPAC, 2016). The effective alkyl nitrate yield for this VOC mixture can be calculated from the relative flux of $NO_3$ reacting with each BVOC (depending on the BVOC mixing ratio and rate coefficient) and the alkyl nitrate yield for each individual BVOC. Note that, in the absence of laboratory investigations, the alkyl nitrate yield from $NO_3$ + myrcene is simply estimated as 50 ± 30 %, in line with other terpenes (IUPAC, 2016). The

final, averaged yield of alkyl-nitrate is $\alpha(NO_3) = (0.41 \pm 0.31)$. The uncertainty we quote is progagated from uncertainty in the rate coefficient and the individual alkyl nitrate yields as reported by IUPAC and also the standard deviation of the mean concentration of the BVOC during the campaign. Clearly, given the large variability in night-time BVOC at this site, uncertainty associated with alkyl nitrate yields and the assumption that all BVOCs with sinificant reactiviy for $NO_3$ were measured, this campaign average value of $\alpha(NO_3)$ should be considered only as a rough indicator. Assuming that each $NO_3$ generated reacts rapidly with a BVOC, the night-time production rate of ANs is then given by:

$$P(\Sigma ANs)_{night} = \alpha(NO_3)k_{(NO_2+O_3)}[NO_2][O_3] \qquad (2)$$

Taking the mean night-time mixing ratios of $[NO_2]$ ($2.7 \pm 2.1$ ppbv) and $[O_3]$ ($45 \pm 11$ ppbv) over the same time period, using the temperature dependent rate coefficient for the reaction between $NO_2$ and $O_3$ ($1.4 \times 10^{-13}$ exp(-2470/T) $cm^3$ molecule$^{-1}$ s$^{-1}$) we calculate the production rate of ANs (Eqn. 2) to be: $P(\Sigma ANs)_{night} \approx 90$ pptv hr$^{-1}$. This analysis implicitly assumes that indirect loss of $NO_3$ via the heterogeneous loss of $N_2O_5$ are insignificant compared to direct loss via reaction with BVOCs. This is expected for a forested region in summer and has been shown to be the case for the Taunus Observatorium (Crowley et al., 2010; Sobanski et al., 2016b) where, except for few occasions when the residual layer is sampled, the $NO_3$ lifetime with respect to gas-phase reactions with BVOCs is too short for indirect, heterogeneous loss to compete unless the mountain-site is in fog.

Given the large uncertainty associated with $\alpha(NO_3)$ and also the variability in $NO_2$ and $O_3$ this average value can be considered consistent with the increase in ANs observed in the two hours following sunset during either the PARADE or NOTOMO campaigns.

### 4.4.1 Day-time generation of ANs in PARADE via OH reactions

To calculate the day-time production of ANs from OH initiated degradation of VOCs, it is necessary to know the OH concentration. During the PARADE campaign, OH was measured on only a few days that did not cover those used to derive our diel profiles. Following Bonn et al. (2014), who performed a detailed analysis of OH measurements and their correlation with $J(O^1D)$ during PARADE, we calculate $[OH] = 1.8 \times 10^{11} \times J(O^1D) \approx 3 \times 10^6$ molecule cm$^{-3}$ for the mean [OH] between 11:00 and 13:00 UTC. Bonn et al. (2014) report a maximum uncertainty of a factor two for [OH] derived in this manner. For an approximate estimate of day-time ANs prodution, we take the campaign mean mixing ratios of each VOC between 11:00 and 13:00 UTC as listed in Table 1. Based on the individual rate coefficients, alkyl nitrate yields and mean, noon-time concentrations of the VOCs measured (also listed in Table 1) and using equation (3);

$$P(\Sigma ANs)_{day} = \Sigma_i \left( \alpha_i(OH)k_{OH+RH_i}[OH][RH_i] \right) \qquad (3)$$

we obtain a noon-time ANs production rate of $P(\Sigma ANs)_{day} \approx 70^{+70}_{-35}$ pptv hr$^{-1}$ where the reported uncertainty is due only to the uncertainty in OH concentrations. As illustrated in Table 1, ~80% of the total, noon-time, $\Sigma$ANs production rate is accounted for by the four biogenic VOCs measured: limonene, myrcene, $\alpha$-pinene, and isoprene. Given that the concentrations of these short lived biogenics are expected to be variable due to the spatial inhomogeneity of emission sources and their dependence on temperature and light levels, the use of campaign averages can provide only a rough indicator of AN production rates. In addition, there is considerable uncertainty associated with the AN yields of the biogenics, which in the absence of measurements, partially stem from structure-reactivity relationships (Perring et al., 2013). The largest uncertainty is however related to the assumption that the reactions of OH are accounted for by the VOCs measured. In forested regions "missing" OH reactivity has been frequently reported (Nölscher et al., 2012; Nolscher et al., 2013), indicating unknown sinks for OH with organic trace gases, which can account for up to 80 % of the observed reactivity. In the case of missing reactivity, the $\Sigma$ANs yields calculated via Eq. (3) are lower limits. Moreover, this expression also neglects the formation of peroxy radicals from the Cl-atom initiated oxidation of VOCs, which may also react with NO to form ANs. As ClNO$_2$ was observed at elevated concentrations on some days during PARADE (Phillips et al., 2012) its main influence on oxidation processes is in the early morning, when OH levels are comparably low. Despite these uncertainties, the calculations above indicate that the value of $P(\Sigma ANs)_{day}$ thus obtained is comparable with the estimated night-time production, which is consistent with the conclusions of (Fry et al., 2013) also made at a forested site with urban influence.

From the discussion above, it is apparent that the relative rates of noon- and night-time generation of $\Sigma$ANs depends on the relative levels of OH and NO$_3$ (a factor $\approx$ 200 in favour of NO$_3$), the yields of ANs (generally larger for NO$_3$) and the rate constant for reaction of OH with VOCs. The large night-time production rate from NO$_3$ degradation of VOCs in this forested enviroment is mainly a consequence of the selective reactivity of NO$_3$ towards terpenes, which have large AN-yields.

Although we calculate similar production rates of $\Sigma$ANs during the night when NO$_3$ is present, the daytime maximum in the $\Sigma$ANs mixing ratio is significantly larger, which has a number of likely causes. The first is related to missing OH reactivity, which, depending on the hydrocarbons involved, could potentially increase the OH-initiated rate of formation of $\Sigma$ANs yield by large factors. For example, if the hydrocarbons we measured would account for only 50 % of the OH reactivity and the missing ones were biogenic in nature (i.e. terpenoids with large AN-yields) we could expect more than a factor of two increase in calculated $P(\Sigma ANs)_{day}$. A further potential cause for larger daytime $\Sigma$ANs mixing ratios is a reduced loss of daytime $\Sigma$ANs with respect to chemical and depositional loss and condensation. This being a consequence of the different chemical composition and volatility of the ANs generated from NO$_3$- compared to OH-initiated oxidation. It is well established in chamber studies that the NO$_3$ induced oxidation of biogenics leads to highly functionalized ANs that partition largely to the aerosol phase and that the NO$_3$ oxidation of biogenic VOCs can lead to appreciable organo-nitrate content in atmospheric particulate matter (Fry et al., 2011; Fry et al., 2014; Boyd et al., 2015). Ambient measurements of aerosol composition show that night-time-generated organic nitrates formed in NO$_3$ + BVOC reactions are efficiently transferred to

the condensed phase (Rollins et al., 2012; Fry et al., 2013; Xu et al., 2015; Kiendler-Scharr et al., 2016), which is confirmed by modelled vapour pressures of the OH- and $NO_3$- initiated organic nitrate products from BVOC oxidation, the latter being substantially lower (Fry et al., 2013).

Table 3 shows that (of the BVOCs measured) limonene accounts for 40 % of the $NO_3$ loss rate, myrcene 30% and $\alpha$-pinene 29 %. Studies of the reaction between $\alpha$-pinene and $NO_3$ show that the yield of secondary organic aerosol (SOA) is less than 10 % (Hallquist et al., 1999; Perraud et al., 2010; Fry et al., 2014; Nah et al., 2016) with reports of the alkyl nitrates formed being exclusively in the gas-phase (Fry et al., 2014). This contrasts strongly with the situation for limonene, where SOA yields of up to $\approx$ 60 % have been reported, with more than 80 % of the alkyl nitrates formed being in the aerosol phase. There is no experimental data on SOA yields in the reaction between $NO_3$ and mycene or on the gas-aerosol partitioning of the alkyl-nitrates formed. Recent experiments on $\beta$-pinene (Boyd et al., 2015) have shown that the SOA yield is neither strongly dependent on the relative concentrations of HO$x$ and NO$x$, which determines the nature of the end-products formed, nor on the relative humidity or seed-aerosol used. If we make the broad assumptions that 1) the SOA yield from mycene is the same as limonene and 2) that the fraction of ANs in the condensed phase is comparable to those found for limonene, we calculate that > 60 % of the alkyl nitrates formed at night at this site will be present in the aerosol phase and thus not detected by our instrument. In other words we would expect equation (3) to yield production rates of $\Sigma$ANs that exceed those derived from gas-phase $\Sigma$AN measurements by a factor between two and three.

We conclude that the apparent lower lifetime of night-time generated $\Sigma$ANs is thus likely to be the result of an increased fraction of low-volatility ANs gormed from terpenes initially reacting with $NO_3$ compared to OH-initiated oxidation, leading to a larger relative rate of SOA formation and partitioning of ANs to the condensed phase. (Fry et al., 2013) have shown that, at an urban / forested site in Colorado, the peak in particle phase organic nitrates occurs at night-time. The condensed phase ANs can undergo hydrolysis to $HNO_3$ (over a period few hours (Lee et al., 2016b)), and thus irreversible loss from the gas-phase, the latter enhanced by the lower temperatures and higher relative humidities encountered at night-time (Hallquist et al., 2009; Lee et al., 2016a).

## 4.5 Effective yield of ANs from correlation between $\Sigma$ANs and $O_3$.

A positive correlation between organic nitrates and $O_3$ has been observed (Kourtidis et al., 1993; Williams et al., 1997; Roberts et al., 1998; Schrimpf et al., 1998; Day et al., 2003) and is due to the common production pathways of these trace gases. In rural and semi-rural locations, the build up of $O_3$ during the day is related to the NO$x$ catalysed photo-oxidation of VOCs, including the reaction of organic peroxy radicals with NO$x$. These processes also dominate the daytime production of organic nitrates. As discussed in Sect. 1, alkyl nitrates are produced via a minor branch of the reaction between NO and organic peroxy radical, while the majority of reactive collisions result in the formation of $NO_2$ and (via its photolysis) the formation of $O_3$ (reaction R4 to R5). Laboratory experiments have shown that the branching ratio to ANs is strongly dependent on the identity of the peroxy radical and also varies with temperature and pressure (Perring et al., 2013). As we

measure total alkyl nitrates, the effective branching ratio is determined by the particular VOC mixture encountered. Following the methodology developed by the Berkeley group (Day et al., 2003; Rosen et al., 2004) the production rate of $O_3$ ($P_{O3}$) is given by eqn. (4).

$$P_{O3} = \Sigma_i\big(\gamma_i(1 - \alpha_i(OH))k_{OH+RH_i}[OH][RH_i]\big)$$
Eq. (4)

where $\alpha(OH)$ is the branching ratio to nitrate formation in the reaction between the OH-generated organic peroxy radical and NO, and $\gamma$ is the number of $O_3$ produced per VOC oxidized, which can be between one and three but is equal to two for many atmospherically relevant VOCs, given sufficient NO (see Table 1) (Rosen et al., 2004). This can be combined with

10   eqn. (3) to give:

$$\frac{\Delta O_3}{\Delta\Sigma ANs} = \frac{\int(P_{O3}-L_{O3}+E_{O3})dt}{\int(P_{\Sigma ANs}-L_{\Sigma ANs}+E_{\Sigma ANs})dt}$$
Eq. (5)

$$\frac{\Delta O_3}{\Delta\Sigma ANs} = \frac{2(1-\alpha)}{\alpha} \approx \frac{2}{\alpha}$$
Eq. (6)

In which $L$ represents loss terms (chemical and deposition) and $E$ represents entrainment, respectively. The ratio of $O_3$ to $\Sigma ANs$ after the OH oxidation of a VOC mixture has proceeded for a certain time, $dt$, is given by Eq. (5). At sufficiently high levels of OH, VOCs and NO, the photochemical production terms can be assumed to be larger than the loss or entrainment terms and Eq. (5) simplifies to Eq. (6). In principal, a plot of [$O_3$] versus [$\Sigma ANs$] should then yield a straight line, with a

slope that is proportional to an average value of the branching ratio to ANs. The average values of $\alpha(OH)$ we calculate using the measurement of $\Sigma ANs$ and $O_3$ is designated $\alpha(OH)_{av}^{\Sigma ANs}$. Alternatively, an average value of $\alpha(OH)$ can be calculated from measurement of the VOCs that react with OH, their rate constant and the individual yield of alkyl nitrate from each reaction and the $O_3$ yield, which we then designate $\alpha(OH)_{av}^{VOCi}$.

For both the PARADE and NOTOMO campaigns, we analysed the mixing ratios of $\Sigma ANs$ and $O_3$ between 11:00 and 13:00

UTC (around the peak in $J(O(^1D))$ and thus OH levels) to calculate $\alpha(OH)_{av}^{\Sigma ANs}$. The results, displayed in Fig. 7, indicate a value of $\alpha(OH)_{av}^{\Sigma ANs}$ (PARADE) $= 7.2 \pm 0.5$ % ($R^2 = 0.49$). In this analysis, the intercept, $28.6 \pm 1.2$ ppbv, may be thought of as the average background level of $O_3$. In the NOTOMO dataset, the yield of $\Sigma ANs$ is low and the data very scattered with a poor correlation coefficient, hence no fit was carried out and we simply plot (black lines) two lines which encompass the whole dataset with corresponding values of $\alpha(OH)_{av}^{\Sigma ANs}$ equal to 0.5 % and 1.7 % which is significantly lower than that

derived for PARADE. For NOTOMO, in which low mixing ratios of $\Sigma ANs$ were encountered, the vertical grouping of the data apparent in Fig. 7 (i.e. low resolution in concentration) is a result of the corrective procedure for extracting mixing ratios from raw data obtained in the hot and cold inlets, which involves iterative numerical simulation which converges when

1% agreement between observation and simulation is achieved. The average correction factor for $\Sigma$ANs during NOTOMO was 1.1 (Sobanski et al., 2016a) and, based on a series of laboratory experiments, the (NO$x$ dependent) uncertainty associated with this factor is expected to be less than 50 %.

For the PARADE campaign, during which VOCs were measured, we derive a value of $\alpha(OH)_{av}^{VOCi} = 6.1$ % for the same period around noon (UTC) which is consistent with the value of $7.2 \pm 0.5$ % derived from measured $\Sigma$ANs and [O$_3$]. Irrespective of method used to derive the branching ratio to $\Sigma$ANs formation, the high values obtained reflect the fact that a significant fraction of OH reactivity is due to biogenic VOCs (especially terpenoids including isoprene, monoterpenes and sesquiterpenes), that have large yields of alkyl nitrates. To emphasise the different efficiency of AN production from VOCs of biogenic origin (BVOC) and anthropogenic origin (AVOC), Table 1 is separated into biogenic and anthropogenic VOCs and from the separate summed $P$(ANs) and $P$(O$_3$) we calculated the effective value of $\alpha(OH)_{av}^{VOCi}$ that would have been obtained considering each class of VOC individually. The values thus obtained are $\alpha(OH)_{av}^{BVOCi} = 17.3$ % and $\alpha(OH)_{av}^{AVOCi} = 1.4$ %. Note that, for this particular hydrocarbon mixture, the O$_3$ production rates are much larger for the AVOCs (~ 1000 compared to ~400 for the BVOCS).

To some extent, the agreement may be fortuitous as both methods to derive $\alpha(OH)_{av}$ involve assumptions that may be only partially applicable. The value of $\alpha(OH)_{av}^{VOCi}$ calculated using individual VOC measurements and their respective alkyl nitrate yields is associated with a rather large uncertainty as it assumes that all VOCs with which OH reacts were actually measured during the campaign and that their alkyl nitrate yields are well known. For example, the products of isoprene oxidation, methyl-vinyl-ketone and methacrolein have high reported $\alpha(OH)$ values (Paulot et al., 2009) and high rate constants for reaction with OH which, if taken into account would increase the value of $\alpha(OH)$. Regarding the individual nitrate yields, some values used to calculate the average (see Table 1) are not precisely determined in the literature while others are estimated. For example, the most recent measurements of the yield of alkyl nitrates formed in the reaction of OH with isoprene in the presence of NO$x$ (one of the best studied reaction systems) ranges from 6 to 13% (Xiong et al., 2015) and remains a source of uncertainty (Perring et al., 2013).

Our calculation of $\alpha(OH)_{av}^{VOCi}$ will also be biased if e.g. local emissions of biogenics are larger than those averaged over the time period over which O$_3$ and ANs were formed. Likewise, neglecting terms for entrainment and loss of ANs and O$_3$ will introduce a variable bias into calculations of $\alpha(OH)_{av}^{\Sigma ANs}$. The rapid loss of multifunctional ANs from terpene oxidation will bias the analysis to low vales of $\alpha(NO_3)$ (Fry et al., 2013). In the absence of information regarding the condensation rate or efficiency of deposition of a mixture of multi-functional nitrates or O$_3$ to the topographically complex terrain at the Taunus Observatory a more detailed analysis is not warranted. The analysis does however make comparison with similar analyses for $\Sigma$ANs measurements possible, and our derived values of $\alpha(OH)_{av}^{\Sigma ANs}$ are consistent with those summarised by Perring et al (Perring et al., 2013) obtained both by observation of $\Sigma$ANs (7.1 % $> \alpha(OH)_{av}^{\Sigma ANs} > 0.8$ % ) and calculated from VOC measurements (10.6 % $> \alpha(OH)_{av}^{VOCi} > 0.1$ %) in various rural and urban locations. A similar analysis by Fry et al. (2013) of

$\Sigma$AN and $O_3$ mixing ratios obtained during summer at a forest site with urban influence resulted in a value of $\alpha(OH)_{av}^{\Sigma ANs} = 2.9\ \%$, intermediate between the value presented here for the PARADE and NOTOMO campaigns.

### 4.5.1 Inter-annual / seasonal differences in $\alpha$(OH), PARADE versus NOTOMO

The difference in $\alpha(OH)_{av}^{\Sigma ANs}$ between the July-2011 PARADE campaign (7.2 ± 0.5 %) and the Aug-Sept. 2015 NOTOMO campaign (< 2 %) is significant and cannot be explained by the uncertainty in the measurements of [$O_3$] or [$\Sigma$ANs] (see section (3.1).

There are a number of potential causes for the apparent difference in $\alpha(OH)_{av}^{\Sigma ANs}$ between the two campaigns. We first consider the validity of the assumption (Eq. 6) that losses and/or entrainment of ANs and $O_3$ can be neglected. Xiong et al.
(2015) have shown that losses of isoprene derived nitrates (IN) due to reaction with OH (Lee et al., 2014) and photolysis can be rapid and start to deplete IN before photochemical production maximises at the peak of the daytime OH-profile. They also indicate that early morning entrainment of IN from the residual layer (where pre-dawn AN mixing ratios may be a factor of 10 larger) can influence its diel profile. This may be particularly relevant for our mountain site, where influence from the free troposphere can be significant. Along with photochemical degradation, dry deposition / hydrolysis can contribute to the
alkyl nitrate sink, especially for those derived from biogenic VOCs (Jacobs et al., 2014; Rindelaub et al., 2015). Xiong et al. (2015), report efficient wet and dry deposition of nitrates derived from isoprene with significantly lower mixing ratios measured in conditions of reduced photochemical reactivity / rain. The restriction of the analysis period to close to the maximum of the OH profile should reduce any bias introduced by the assumptions inherent to Eq. 6, but will not remove it totally. As the lifetime with respect to chemical loss/ deposition of alkyl nitrates derived from biogenic VOCs is expected to
be shorter than that of $O_3$, the sampling of progressively aged air masses will bias $\alpha(OH)_{av}^{\Sigma ANs}$ to low values when calculated from $O_3$ / $\Sigma$AN correlations. A low value of $\alpha(OH)_{av}^{\Sigma ANs}$ during NOTOMO could conceivably be the result of sampling on average older air masses than during PARADE. The lower NO$x$ levels in NOTOMO would support this contention, though in the absence of NO$y$ measurements, is not conclusive.

A further, related explanation for low values of $\alpha(OH)_{av}^{\Sigma ANs}$ during NOTOMO is that the average lifetime of $\Sigma$ANs was
shorter than during PARADE, due e.g. to chemically distinct ANs being generated, this resulting from a different hydrocarbon mix present during the campaigns. As no BVOC measurements were taken during NOTOMO we can only speculate on potential reasons for this. We first note that the campaigns were in different seasons and propose that the mountainside vegetation was in different (seasonal) growth phases as NOTOMO (2015) took place during July, and was characterised by recurrent damp and foggy conditions whereas the PARADE campaign (2011) took place later in the year
(mid-August to mid-September) during the transition from summer to autumn. The plant-physiology controlled BVOC emissions depend not only on the temperature and insolation during the two campaigns, which were comparable, but also on the weather (temperature, rainfall etc.) during the preceding months, which displays a large inter-annual variability at this mountain site. For example, a switch from α-pinene dominant to β-pinene or limonene dominant emissions from the mixed

vegetation could influence not only the ΣANs production rate (there is considerable uncertainty associated with the AN yields, see above) but also the degree to which the ANs are transferred to the particle-phase as evidenced by the different yields of secondary organic aerosol formed in these systems (Mutzel et al., 2016). Measurements of OH reactivity at this site in 2011 indicate seasonal differences in the production and emission of BVOCs and suggest that unmeasured primary

biogenic emissions contribute significantly to the observed OH reactivity, especially in late summer (PARADE) (Nolscher et al., 2013).

Given the mixed forest / urban location, the hydrocarbon mixture can also be influenced by different, average contributions from anthropogenic emissions. An increase in the relative abundance of anthropogenic to biogenic VOCs during NOTOMO would decrease the value of $\alpha(\text{OH})_{av}^{\Sigma \text{ANs}}$ (see above) and thus reduce the production rate of ΣANs. If a substantial

anthropogenic contribution to the VOC mixture was indeed present during NOTOMO, a further reduction in the apparent NOTOMO yield of ANs compared to PARADE could result from increased rates of deposition of the ANs of anthropogenic origin, which are smaller and have more oxidised functional groups per carbon and should thus be more hydroscopic (Fry et al., 2013).

**4. Conclusions**

By measuring total organic nitrates, NO, $NO_2$ and $NO_3$ we have shown that a significant fraction (up to 75 %) of $NO_x$ is sequestered as gas-phase organic nitrates at this forested site with urban influence. During the Aug-Sept. 2011 PARADE campaign, ΣANs and ΣPNs were measured in similar concentrations, whereas in NOTOMO (July 2015) formation of ΣANs was weaker. The difference between the years / seasons may be due to several factors including varied overal rates of

20 BVOC emission and BVOC speciation during the two campaigns and also breakdown of assumptions used to calculate the effective ΣAN yield. Based on an estimate of the OH concentration and the $NO_3$ production term we show that both night-time ($NO_3$ initiated) and daytime (OH initiated) chemistry contributes to the formation of ANs, which is reflected in their diel profile. Daytime ANs are more abundant, possibly reflecting their lower rates of loss, a result of differing night-time versus daytime boundary layer meteorology / dynamics and chemical properties of the ANs formed.

 **Acknowledgements**

We thank Heinz Bingemer for logistical support and use of the facilities at the Taunus Observatory during the NOTOMO and PARADE campaigns. We thank Horst Fischer for making the PARADE NO dataset available and the HLNUG for provision of meteorological data. We thank DuPont for provision of a sample of the FEP TE 9568 suspension used to coat

the cavity walls. This work was carried out in part fulfilment of the PhD of Nicolas Sobanski at the Johannes Gutenberg University in Mainz, Germany.

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

**Table 1.** VOCs measured during the PARADE campaign by GC-MS/FID

| VOC | Mean noon-time mixing ratio (pptv) | $k_{OH + VOC}$ ($cm^3$ molecule$^{-1}$) | $\alpha$(OH) | P(ANs) (pptv hr$^{-1}$) | P(O$_3$) (pptv hr$^{-1}$) |
|---|---|---|---|---|---|
| limonene | 28 | $1.71 \times 10^{-10}$ | 0.23 | 11.89 | 74.5 |
| myrcene | 20 | $2.15 \times 10^{-10}$ | 0.23 | 10.68 | 76.2 |
| $\alpha$-pinene | 49 | $5.37 \times 10^{-11}$ | 0.18 | 5.12 | 66.4[a] |
| isoprene | 85 | $1.01 \times 10^{-10}$ | 0.07 | 6.49 | 177.3 |
| *sum BVOCs* | | | | 34.18 | 394.3 |
| | | | | | |
| ethylbenzene | 26 | $7.10 \times 10^{-11}$ | 0.072 | 1.44 | 36 |
| *n*-Pentane | 211 | $3.9 \times 10^{-12}$ | 0.105 | 0.93 | 23 [a] |
| *i*-pentane | 282 | $3.9 \times 10^{-12}$ | 0.07 | 0.83 | 22[a] |
| p-xylene | 46 | $1.43 \times 10^{-11}$ | 0.097 | 0.69 | 13 |
| *n*-Butane | 265 | $2.54 \times 10^{-12}$ | 0.077 | 0.56 | 19 [a] |
| hexane | 56 | $5.61 \times 10^{-12}$ | 0.141 | 0.56 | 8.3 [a] |
| m-xylene | 46 | $1.43 \times 10^{-11}$ | 0.074 | 0.53 | 13 |
| propene | 105 | $2.63 \times 10^{-11}$ | 0.015 | 0.45 | 58 |
| benzene | 78 | $1.23 \times 10^{-11}$ | 0.034 | 0.35 | 19 |
| *i*-Butane | 131 | $2.33 \times 10^{-112}$ | 0.096 | 0.32 | 7 [a] |
| cis-2-Butene | 15 | $5.6 \times 10^{-11}$ | 0.034 | 0.31 | 17 |
| o-xylene | 21 | $1.37 \times 10^{-11}$ | 0.081 | 0.25 | 5.8 |
| toluene | 126 | $6.0 \times 10^{-12}$ | 0.029 | 0.24 | 15 |
| ethene | 246 | $8.5 \times 10^{-12}$ | 0.0086 | 0.19 | 45 |
| propane | 330 | $1.15 \times 10^{-12}$ | 0.036 | 0.15 | 7.9 |
| methane | $1.8 \times 10^6$ | $6.9 \times 10^{-15}$ | 0.0005 | 0.07 | 268 |
| ethane | 590 | $2.6 \times 10^{-13}$ | 0.019 | 0.033 | 3.3 |
| HCHO | 1940 | $8.5 \times 10^{-12}$ | 0 | 0 | 358 |
| CO | $1.2 \times 10^5$ | $2.4 \times 10^{-13}$ | 0 | 0 | 303 |
| *sum AVOCs* | | | | 7.9 | 1071.1 |
| | | | | | |
| **Sum** | | | | **42.1** | **1465.4** |

BVOCs (mainly biogenically emitted VOCs) blue text, AVOCs (mainly anthropogenically emitted VOCs) in black text. The production rate of alkyl nitrates and O$_3$ are calculated based on an OH concentration of 3 x 10$^6$ molecule cm$^{-3}$. Values of $\alpha$(OH) were taken from Perring et al. (2013), values of $k_{OH+VOC}$ were taken from Atkinson and Arey (2003a). [a]The number of ozone molecules produced per VOC oxidized is 2.85 (all others are 2) (Rosen et al., 2004).

**Table 2.** Minimum, mean and maximum values for relative humidity, temperature, ozone, NO, $NO_2$ and organic nitrates during the PARADE and NOTOMO campaigns.

| Species | PARADE (Aug.-Sept. 2011) | | | NOTOMO (Jul. 2015) | | |
|---|---|---|---|---|---|---|
| | Minimum | Maximum | Mean | Minimum | Maximum | Mean |
| Relative humidity (%) | 38 | 100 | 77 | 27 | 100 | 70 |
| Temperature (°C) | 6 | 27 | 15 | 6 | 33 | 17 |
| $O_3$ (ppbv) | 8 | 81 | 41 | 17 | 150 | 48 |
| $NO_2$ (ppbv) | 0.3 | 21 | 2.7 | 0.1 | 15 | 2 |
| NO (ppbv) | < LOD | 5 | 0.3 | 0[a] | 3[a] | 0.3[a] |
| ΣPNs (ppbv) | < LOD | 2 | 0.5 | < LOD | 3.2 | 0.7 |
| ΣANs (ppbv) | < LOD | 1.2 | 0.3 | < LOD | 0.8 | 0.1 |

[a] calculated using measurements of $NO_2$, $O_3$ and $J(O^1D)$ and assuming photo-stationary state (see text for details).

**Table 3.** Mean, night-time BVOC mixing ratios during PARADE.

| BVOC | Mean $\pm$ SD[1] | $k_{NO3 + VOC}$ [2] | Relative flux | $\alpha_{NO3 + VOC}$ |
|---|---|---|---|---|
| limonene | $23.9 \pm 12$ | $120 \pm 36$ | $0.40 \pm 0.23$ | $0.5 \pm 0.2$ |
| myrcene | $19.7 \pm 11.8$ | $110 \pm 33$ | $0.30 \pm 0.20$ | $0.5 \pm 0.3$ [3] |
| $\alpha$-pinene | $33.1 \pm 14.1$ | $62 \pm 16$ | $0.29 \pm 0.14$ | $0.17 \pm 0.3$ |
| isoprene | $14.0 \pm 14.3$ | $6.5 \pm 2.6$ | $0.01 \pm 0.01$ | $0.75 \pm 0.15$ |

[1]Mixing ratios in pptv. [2]Rate constants (298 K) in $10^{-13}$ cm$^3$ molecule$^{-1}$ s$^{-1}$. Rate constants (and associated uncertainty) and organic nitrate yields (with spread of measurements) were taken from IUPAC (IUPAC, 2016). [3]In the absence of experimental data, this value is an estimate with expanded eror limits.

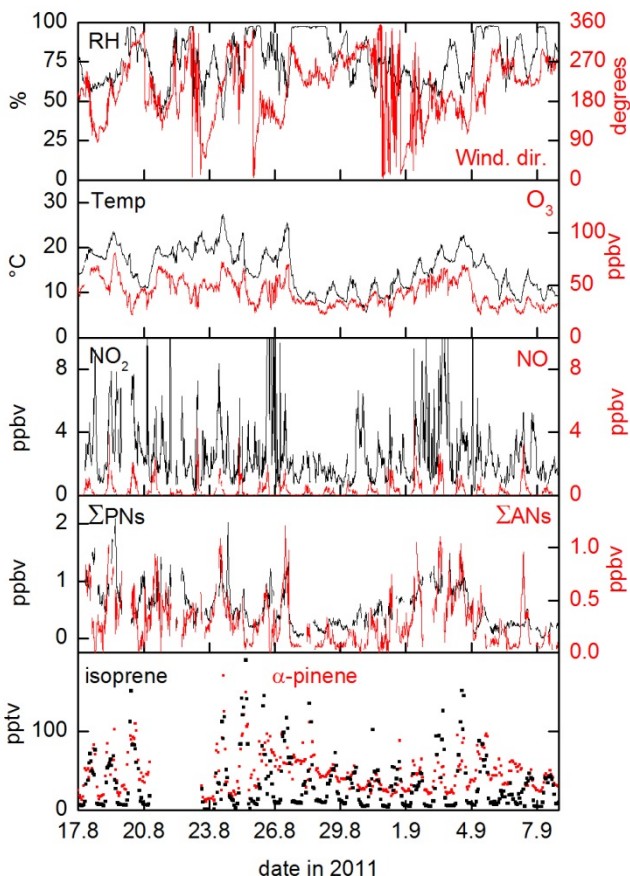

**Fig. 1.** PARADE 2011: Time-series of relative humidity (RH), wind direction, temperature, and the $O_3$, $NO_2$, NO, $\Sigma$PNs, and $\Sigma$ANs mixing ratios. The lowest panel shows the mixing ratios of isoprene (black datapoints) and $\alpha$-pinene (red datapoints).

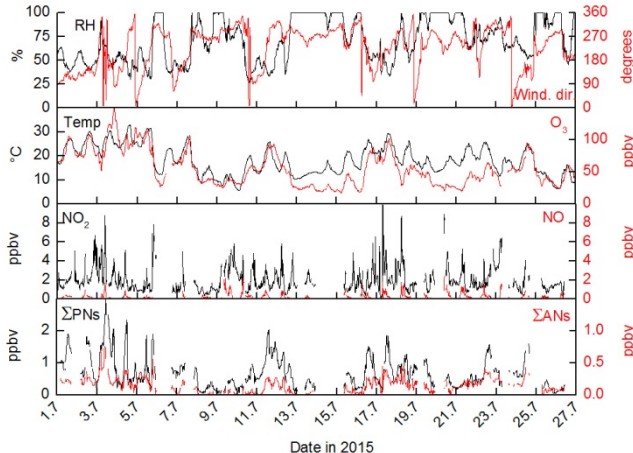

**Fig. 2.** NOTOMO 2015: Time-series of relative humidity (RH), wind direction, temperature and the mixing ratios of O$_3$, NO$_2$, NO, ΣPNs, and ΣANs. The NO mixing ratios were calculated assuming photo-stationary state as described in the text.

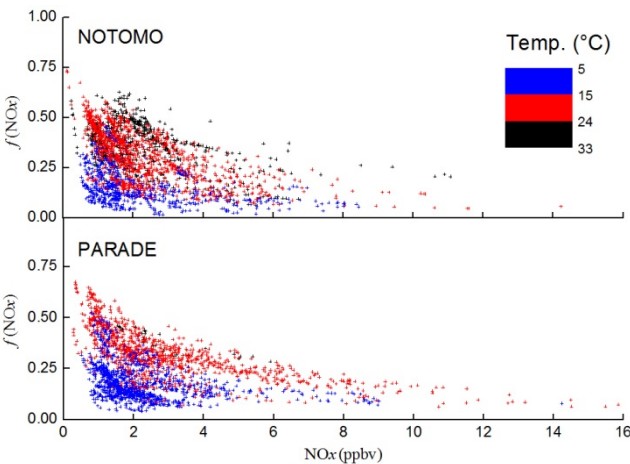

**Fig. 3.** The fraction of NO$x$ sequestered as organic nitrates $f(\text{NO}x)=$ [ΣPNs]+[ΣANs] /([ΣPNs]+[ΣANs]+[NO$x$]) as a function of NO$x$ for the NOTOMO and PARADE campaigns, both colour coded for temperature.

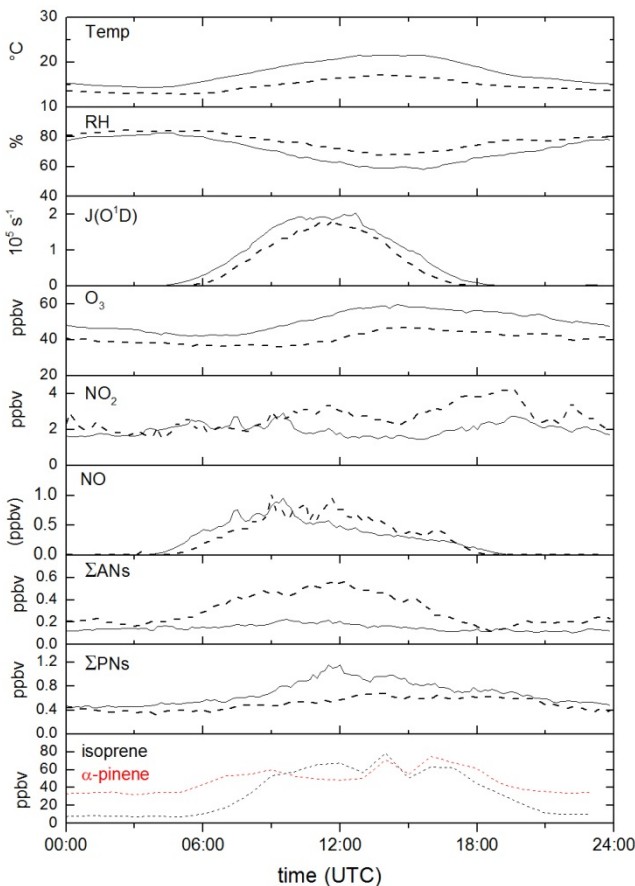

**Fig. 4.** Mean diel profiles during PARADE (dashed lines) and NOTOMO (slid lines) of J(O¹D), [O₃], temperature, [NO₂], [ΣPNs] and [ΣANs]. Isoprene and α-pinene were measured only during PARADE.

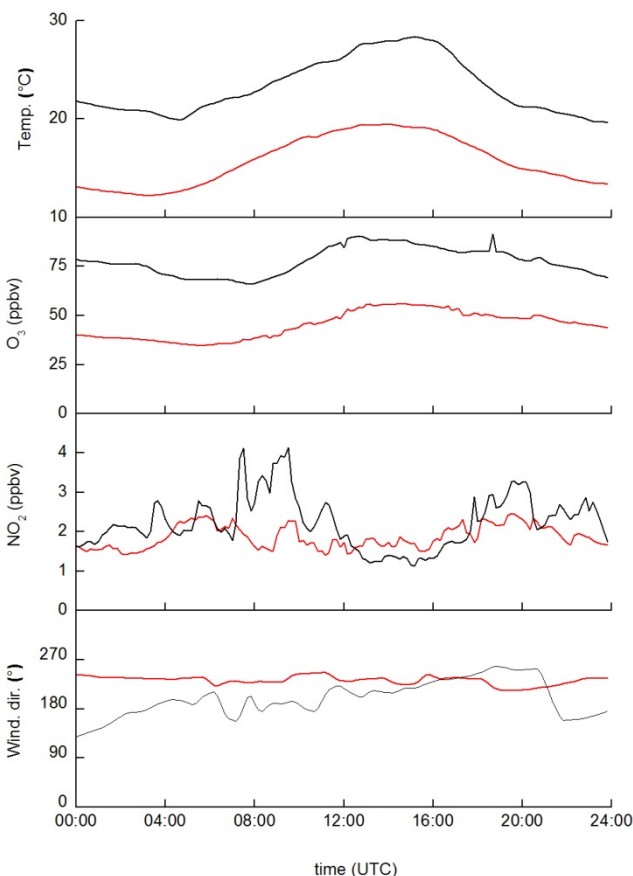

**Fig. 5.** Mean daily profiles of [$O_3$], temperature, [$NO_2$], and wind direction for the NOTOMO campaign separated into days with a clear influence from local rush-hour traffic (black). The rest are shown in red.

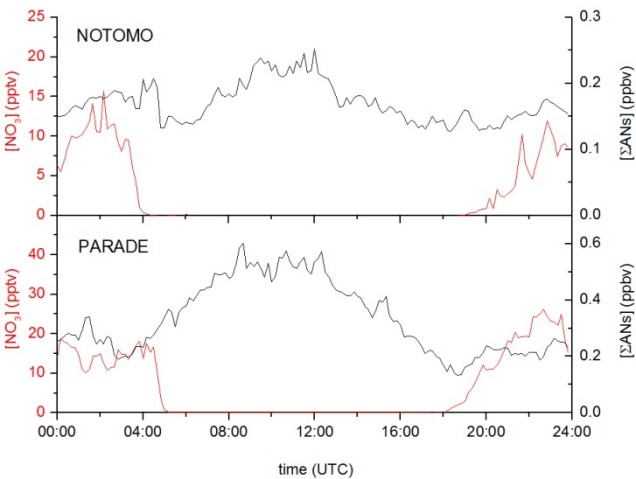

**Fig. 6.** Mean daily profiles of [NO$_3$] (red) and [ΣANs] (black) for the NOTOMO (top) and PARADE campaigns (bottom) for relative humidity < 90%.

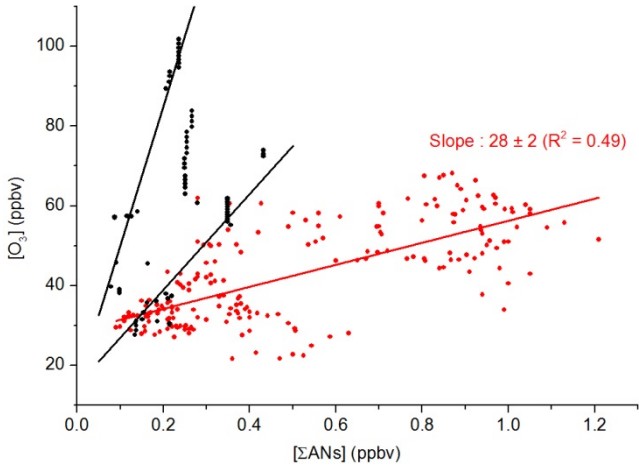

**Fig. 7.** [O₃] versus [ΣANs] measured between 11:00 and 13:00 UTC during PARADE (red data points) and NOTOMO (black data points). The red line is the best fit for PARADE. For NOTOMO the black line are chosen to encompass all possible values of α assuming a background O₃ level of 15 ppbv. The apparent, poor resolution in the NOTOMO data is due to the low yield and the iterative correction procedure as described in the text.