# Peer review of "Day- and Night-time Formation of Organic Nitrates at a Forested Mountain-site in South West Germany"

_Atmospheric Chemistry and Physics, 2016_

## Referee Comment (RC1) · Anonymous Referee #1 · 7 Dec 2016

Sobanski et al. present analysis of the measurements of organic nitrates from two field deployments at the Taunus Observatory in Germany. This analysis is a useful contribution to our understanding of the role of organic nitrates in the NOx budget on a global scale, and raises interesting questions about the relative fate of organic nitrates during the day and night. I would suggest publication after the following comments are addressed.

General Comments:

1. When calculating average production rates of alkyl nitrates during day and night, the authors use campaign average values for each term. Given the variety of conditions sampled during the campaign, it seems possible that using campaign averages will bias

the results (if, for example, the mixture of VOCs and therefore the alkyl nitrate yield is different on nights with higher concentrations of NO2 and O3). The use of campaign average values in calculations should either be avoided or the consequences of them discussed.

2. The discussion of the differences in alkyl nitrate yield between PARADE and NO-TOMO should be expanded further. While the proposed explanation, that BVOC emissions were lower during NOTOMO, is plausible, I would appreciate further discussion of alternative explanations for the observations. In particular, the authors should consider the possibility that the NOTOMO observations of O3 and ANs represent a highly aged airmass where the assumptions required for Eq. 6 do not apply.

Specific Comments:

Page 1, Line 26-27: Since HNO3 does not appreciably return NOx to the atmosphere, it is incorrect to describe NOx as being temporarily sequestered as HNO3.

Page 3, Line 3: NOy should be defined in this manuscript.

Page 3, Line 10: The abbreviation TD-CRDS should be defined here, the first time it is used, rather than on page 4.

Page 4, Section 3.1: Is the TD-CRDS measurement of ANs gas-phase only? Given the potential importance of particle-phase chemistry to understanding the nighttime concentration of ANs, the response of the TD-CRDS instrument to particulate organic nitrates should be discussed in this section.

Page 5, Line 25: Was J(NO2) measured directly, or was it modeled?

Page 9, Line 22: Given that direct measurements of NO3 are available for this campaign, why are those measurements not used to calculate the nighttime alkyl nitrate production rate?

Page 9, Line 24 and Line 29: Which days were included when calculating mean nighttime mixing ratios? All days, only days including in Fig. 6, or some other combination?

Page 9, Line 25: A citation for these alkyl nitrate yields should be given.

Page 9, Line 27-28: Some justification for assuming that heterogeneous N2O5 loss is minimal should be included.

Page 9, Equation 2: See general comment 1.

Page 10, Line 5: What uncertainty in the calculated value of OH does this correlation introduce?

Page 10, Line 6-7: Is the campaign mean calculated for all days or for only times included in Fig. 6?

Page 10, Line 10, Equation 3: See general comment 1.

Page 10, Line 20: I typically think of deposition dropping to near zero at night, since turbulent mixing is low. The authors should discuss further the likelihood of enhanced nighttime deposition.

Page 10, Line 23-24: Can the SOA yields reported by Fry et al. 2011, 2014 be used to estimate the fraction of ANs produced that are likely to remain in the gas phase, and can that fraction be used to adjust Eq. 2 to describe only the gas-phase production of alkyl nitrates?

Page 11, Equation 5: The concentration of ozone and ANs should include the effect of chemical loss.

Page 12, Line 3-7: Under the conditions of the NOTOMO campaign, what uncertainty in AN concentration does the correction procedure introduce?

Page 12, Line 20-22: This is likely an overestimate of the range of isoprene alkyl nitrate yield. Recent work on the isoprene branching ratio has generally found branching ratios on the higher end of this range (9-15%) (Teng et al., 2015, Xiong et al., 2015)

Page 12, Line 33: Photolysis and chemical loss of alkyl nitrates is often a more important loss process than deposition (Xiong et al., 2015).

Page 13, Line 6-9: Any explanation for the low concentrations of alkyl nitrates should also be able to explain the high concentration of ozone encountered during the NOTOMO campaign.

Page 13 Line 7: Given that on average, NOTOMO was warmer and sunnier than PARADE (page 8), what magnitude of changes in VOC emissions is expected between the two campaigns? Is this change large enough to explain the low observed yield of alkyl nitrates?

Page 13 Line 8: Based on the mixture of non-biogenic VOCs measured during PARADE, would decreased concentrations of BVOCs lead to a lower average value of alpha? Is this value low enough to explain the observed O3-AN slope during NOTOMO?

Page 18, Line 20, Table 1: Several of the values in this table disagree with those listed in Perring et al, 2013, the listed source for the yield. This includes i-pentane(0.35/0.07), isoprene (0.044/0.07) and i-butane(0.255/0.096). The values in Table 1 should either be updated or new references given.

Technical Corrections:

Page 1 Line 12: "Futher" should be "Further"

Page 6, Line 15: Extra period after Fig. 2.

Page 11, Line 14 Equation 4: There are some mis-matched parentheses in this equation

Page 18, Table 1: The mean noon time mixing ratio unit appears to be pptv, and not ppbv as written

Additional References:

Xiong, F., McAvey, K. M., Pratt, K. A., Groff, C. J., Hostetler, M. A., Lipton, M. A., Starn, T. K., Seeley, J. V., Bertman, S. B., Teng, A. P., Crounse, J. D., Nguyen, T. B., Wennberg, P. O., Misztal, P. K., Goldstein, A. H., Guenther, A. B., Koss, A. R., Olson, K. F., de Gouw, J. A., Baumann, K., Edgerton, E. S., Feiner, P. A., Zhang, L., Miller, D. O., Brune, W. H., and Shepson, P. B.: Observation of isoprene hydroxynitrates in the southeastern United States and implications for the fate of NOx, Atmos. Chem. Phys., 15, 11257-11272, doi:10.5194/acp-15-11257-2015, 2015.

Teng, A. P., Crounse, J. D., Lee, L., St. Clair, J. M., Cohen, R. C., and Wennberg, P. O.: Hydroxy nitrate production in the OH-initiated oxidation of alkenes, Atmos. Chem. Phys., 15, 4297-4316, doi:10.5194/acp-15-4297-2015, 2015.

---

## Referee Comment (RC2) · Anonymous Referee #2 · 9 Dec 2016

**Review of Sobanski et al, "Day- and Night-time Formation of Organic Nitrates at a Forested Mountain site in South West Germany"**

This paper presents two summer field campaigns (2011 and 2015) of organic nitrate data at an urban-influenced mountain site, focusing on meausrements of alkyl nitrate formation and implicatoins for production of ozone, and measurements of peroxynitrates, both using a thermal dissociation – cavity ringdown instrument, which detects these species via thermal dissociation and then detection of $NO_2$. Reference is made to an earlier instrument paper for the measurement methdology and data corrections to account for some known interferences, the focus in this paper is in interpreting the field observations, including comparing across two years with different meteorological conditions. This paper presents a novel combined dataset that will be of interest to the atmospheric chemistry community. I do see a few opportunities to extend the analysis and to compare to additional available measurements, which I mention below, and recommend publication after these revisions.

**Major suggestions:**

1) There are a few additional studies that I would suggest citing to inform your analysis, and to enable comparisons with your data:

- Kiendler-Scarr et al (2016) have just published a series of measurements of aerosol-phase $RONO_2$ around Europe using AMS, and they comment on the ubiquity of $NO_3$ sourced nitrates – this would be a good point of comparison. ("Organic nitrates from night-time chemistry are ubiquitous in the European submicron aerosol," Geophys Res Lett, 10.1002/2016GL069239, 2016.)
- Fry et al. (2013) measured $\Sigma PNs$ and $\Sigma ANs$ at an urban-influenced Colorado site, also in summertime, including doing the same $O_3$ / ANs slope analysis to assess nitrate branching and relevance to $O_3$ formation. Compare yield and VOC mix? ("Observations of gas- and aerosol-phase organic nitrates at BEACHON-RoMBAS 2011," Atmos. Chem. Phys., 13, 8585-8605, 2013.)
- The Ng et al. 2016 review paper that you cite discussed some observations and general trends about organonitrate losses, including hydrolysis, that would be valuable to consider in the context of your claims that nighttime nitrates may be lost more rapidly under damp and foggy conditions during NOTOMO, in contrast to the first year. There are several primary papers cited in Ng that might be relevant here, for example Boyd et al 2015 ("Secondary organic aerosol formation from the β-pinene+NO3 system: effect of humidity and peroxy radical fate," Atmos. Chem. Phys., 15, 7497-7522, 10.5194/acp-15- 7497-2015, 2015.) – but in general I think the conclusions here would be at odds with your observations: nighttime $NO_3$+terpene produced organonitrates are expected to be mostly primary and secondary nitrates, which appear to have longer lifetimes, not shorter, than daytime produced OH-inititated nitrates (which would include more tertiary nitrates)

2) In my view, the weakest point in the paper is the explanation of the difference in PAN/ AN ratio and apparent $O_3$ AN braching ratio across the two years. It would be great to find more evidence to support and interpret this difference. For example:

- even though you don't have VOC measurements in year 2 (bummer!), could you look at e.g. termperature / sunshine differences correlated with VOCs measured within

the PARADE period where you DO have the GCs running, and then extrapolate to the conditions during the second year of measurements?

- Or even use any other GC data taken at that site, whenever, to be able to say something about the potential range of year- over- year variability?
- Can you find literature to point to on how oxidized VOCs like nitrates deposition depends on met conditions (your claim at the top of p. 13)?
- Does the NO: $NO_2$ ratio during the two years support the apparent differences in PAN vs. ANs formation rate?
- Can you find any NOx emissions data or traffic counts or similar to suggest that the NOy mix arriving at the site might be different across the 2 years?

3) Figure suggestions

- It would be valuable to see some of the VOC variability in addition to reporting the mean noontime values in table 1. Could you add the reactively most important VOC or two to Fig. 1, to enable readers to see whether periods of high ANs/PNs correlate with higher VOC? Also, suggest to add the diurnally averaged versoin to Fig. 4 as well. Are all daytime-peaking or some nighttime? Could target trying to ID the dominant $NO_3$+BVOC source of organonitrates at night vs. datytime RO2+NO source, which will help you put the ideas about hydrolysis lifetime and it's structure dependence in context.
- On Fig. 3, can you format the points so they don't obscure one another? it looks like the black points are behind the red, so it's hard to see their spread. Maybe use "+"s instead? Or bin /average data so there aren't so many points on the plot?
- Please "squish" Fig. 4 and 5 on the horizontal axis (or equivalently, make them taller) so they are the same width as Fig. 6, where the diunral pattern is easier to see because of the larger height to width aspect ratio
- Suggest to rethink color scheme on figures. Red/black don't always means the same thing, leading to confusion. For example, could do dots vs solid for years, consistently, and always use color to refer to left/right axis?
- Suggest to add NO to figure 4.
- In caption to Fig. 5, briefly described how you separate out the rush-hour influenced days
- Fig. 7: how did you choose 11-13 UTC for the $O_3$ vs ANs slopes? Did you check consistency using different time periods? Also, could the iterative correction procedure that makes the ANs data look binned on fig. 7 be the reason for lower ANs concentration measurements, too?  What is the relative error on these measurements in each campaign, based on the correction procedure? Could you put error bars on these plots? (Again, might be best to bin first to avoid having a too-busy plot)

**Minor or technical edits:**

1) p. 2 line 1: "during the night (R6) (see below) to produce peroxy radicals which subsequently produce stable organonitrates by any radical terminal reaction. Organic peroxy radicals are also formed in…"

2) p. 2 lin 12 "ultimate to $O_3$ formation."

3) p. 2 line 31-32: suggest to include chemical formulae for each PAN, PPN, MPAN, analogous to how you show $RC(O)O2NO_2$ on line 5 of this page.

4) p 3 line 4: "first measurements of total $\Sigma PNs$ and …"

5) p 3 line 32: clarify that the long observed $NO_3$ lifetime here is presumed due to low VOC mising ratio – correct? If so, could you note the mixing ratio compared to another time where you're not sampling the residual layer?

6) p 5 line 32 what does "HLUG" mean?

7) p. 6 line 10-11. This is uncelar. Add symbols to help reader correlate to table 1, e.g. "alky nitrate yields (alpha(ANs) and production rate of alkyl nitrates (P(ANs)) and ozone (P($O_3$)). The phrase "for ANs and PNs, respectively" I don't understand – P($O_3$) is just based on ANs – what do you mean by the PNs part?

8) p. 6 line 16. HLUG again - ?? and typo in "summarizes"

9) suggest to find/replace "see later" and change to "see below", or vice versa, to make all consistent.

10) p. 7 line 6: Maybe check if Lee et al's SOAS paper also does the sumANS / individual ANs comparison that Beaver's paper did? I think in the supplemental: http://www.pnas.org/content/113/6/1516

11) p. 7 line 16-18: is this a chemical clock? Can you correlate with wind speed?

12) p. 7 line 22: "this site. Furthermore, the temperature dependence suggest that .." (because it's the T-dependent that suggests this, right? Not the f(NOx) values). Also, the last sentenc of this paragraph offers an alternative explanation, so maybe start with "However, note that low temperatures also increase …" and then put some more evidence for one of the other potential explanations, from Major Sugestions #1 above, below in this section.

13) p. 7 line 31: typo "emissions"

14) p. 8 lin 1: unclear. do you mean the mean and daily maxima of each variable? reword.

15) p. 9 line 4: give number for the PARADE campaign too to compare.

16) p. 9 line 15: typo "[$NO_3$] and"

17) p. 9 line 25-26: are you accounting for differences in rate here too, or assuming there is abundant $NO_3$ to fully oxidize all VOCs? Elaborate.

18) p. 10 line 15 typo "environment"

19) p. 10 line 23: SOA yields could be even higher – see summary table in Ng 2016 paper.

20) p. 11 line 24: "than the loss terms, D and E, and Eq. 5 …"

---

## Author Comment (AC1) · 7 Feb 2017

**Referee 1**

In the following, the referee's comments are reproduced (black) along with our replies (blue) and changes made to the text (red) in the revised manuscript.

**General statement:**

Sobanski et al. present analysis of the measurements of organic nitrates from two field deployments at the Taunus Observatory in Germany. This analysis is a useful contribution to our understanding of the role of organic nitrates in the NOx budget on a global scale, and raises interesting questions about the relative fate of organic nitrates during the day and night. I would suggest publication after the following comments are addressed.
We thank the referee for this informed and thorough review and overall positive assessment of our manuscript. The manuscript has been improved in line with the comments listed below.

**General comments**:

1. When calculating average production rates of alkyl nitrates during day and night, the authors use campaign average values for each term. Given the variety of conditions sampled during the campaign, it seems possible that using campaign averages will bias the results (if, for example, the mixture of VOCs and therefore the alkyl nitrate yield is different on nights with higher concentrations of NO2 and O3). The use of campaign average values in calculations should either be avoided or the consequences of them discussed.
The referee refers to use of equations (1) and (2). We appreciate that using campaign averages in these calculations will result in inclusion of data that cover a large spread in conditions and thus production rates. As far as equation (2) is concerned (night-time production), this variability can be expressed by considering the standard deviation in the $NO_2$ and $O_3$ concentrations and the mixing ratios of the BVOCs. The overall uncertainty in the production rate depends on the uncertainty in the branching ratio to ANs and the rate coefficient for the reaction between $NO_2$ and $O_3$. We have added a Table (3) with details about the concentrations and added more information about the calculation to derive the effective campaign averaged, branching ratio and the uncertainty in the ANs branching ratio. We have modified the text as follows: "To estimate the night time-production of ANs, we consider the reaction between $NO_2$ and $O_3$ to be the only $NO_3$ precursor. Of the measured VOCs, isoprene, α-pinene, myrcene and limonene account for > 95% of the $NO_3$ reactivity and have substantial yields of ANs. The mean night-time mixing ratios for these four compounds during PARADE (excluding data where RH > 92 %) are listed in Table 3 along with the corresponding $NO_3$-reaction rate coefficients (*k*) and the alkyl nitrate yields (α) as reported by the IUPAC panel (IUPAC, 2016). The effective alkyl nitrate yield for this VOC mixture can be calculated from the relative flux of $NO_3$ reacting with each BVOC (depending on the BVOC mixing ratio and rate coefficient) and the alkyl nitrate yield for each individual BVOC. Note that, in the absence of laboratory investigations, the alkyl nitrate yield from $NO_3$ + myrcene is simply estimated as 50 ± 30 %, in line with other terpenes (IUPAC, 2016). The final, averaged yield of alkyl-nitrate is α($NO_3$) = (0.41 ± 0.31). The uncertainty we quote is progagated from uncertainty in the rate coefficient and the individual alkyl nitrate yields as reported by IUPAC and also the standard deviation of the mean concentration of the BVOC during the campaign. Clearly, given the large variability in night-time BVOC at this site, uncertainty associated with alkyl nitrate yields and the assumption that all BVOCs with sinificant reactiviy for $NO_3$ were measured, this campaign average value of α($NO_3$) should be

[revised manuscript text omitted]

2. The discussion of the differences in alkyl nitrate yield between PARADE and NOTOMO should be expanded further. While the proposed explanation, that BVOC emissions were lower during NOTOMO, is plausible, I would appreciate further discussion of alternative explanations for the observations. In particular, the authors should consider the possibility that the NOTOMO observations of O3 and ANs represent a highly aged airmass where the assumptions required for Eq. 6 do not apply.

We have extended the discussion in a new section (4.5.1) with the following text

[revised manuscript text omitted]

**Specific Comments:**

Page 1, Line 26-27: Since $HNO_3$ does not appreciably return NOx to the atmosphere, it is incorrect to describe NOx as being temporarily sequestered as $HNO_3$.
We have corrected the text, which now reads: "In the troposphere, a significant amount of NO$x$ can be temporarily sequestered as organic nitrates."

Page 3, Line 3: NOy should be defined in this manuscript.
We now write: "Early attempts to compare total reactive nitrogen NO$y$ (where NO$y$ = NO$x$ + $RO_2NO_2$ + $RONO_2$ + $HNO_3$ + HONO + +) with the sum of individually measured species (Fahey et al., 1986; Buhr et al., 1990; Ridley et al., 1990) revealed that a substantial fraction of NO$y$ was missing."

Page 3, Line 10: The abbreviation TD-CRDS should be defined here, the first time it is used, rather than on page 4.
We now define TD-CRDS on page 3: "We present here an analysis of organic nitrates and $NO_2$ measured using Thermal Dissociation Cavity Ring-Down Spectroscopy (TD-CRDS) during two field campaigns that took place at a forested, semi-rural mountain site in South-Western Germany."

Page 4, Section 3.1: Is the TD-CRDS measurement of ANs gas-phase only? Given the potential importance of particle-phase chemistry to understanding the nighttime concentration of ANs, the response of the TD-CRDS instrument to particulate organic nitrates should be discussed in this section.
The sampled air is drawn through Teflon membrane filters which prevent detection of particulate nitrate (as $NO_2$) following passage through the TD-regions. Text has been added (section 3) to illustrate this: "$NO_2$ and total gas-phase organic nitrates were measured during

both campaigns by TD-CRDS. Membrane filters were used to prevent aerosol from entering the CRD inlets, which would lead to severe reductions in the detection limit, degradation of the cavity mirrors and also to the detection of particulate nitrate (both organic and inorganic) in the TD channels."

Page 5, Line 25: Was J(NO2) measured directly, or was it modeled?
J($NO_2$) was measured. The sentence now reads: "……where $J(NO_2)$ is the photolysis frequency of $NO_2$ (measured using a METCON spectral radiometer) and $k_{(NO+O3)}$ is the rate constant for reaction of NO with $O_3$."

Page 9, Line 22: Given that direct measurements of $NO_3$ are available for this campaign, why are those measurements not used to calculate the nighttime alkyl nitrate production rate?
The AN production term is high even when $NO_3$ concentrations are "zero" or below the detection limit of the instrument (e.g. because BVOC levels are high). As long as reaction with VOCS is the dominant loss process for $NO_3$, it is appropriate to use the $NO_2$ and $O_3$ concentrations. It is mathematically equivalent to using $NO_3$ concentrations if $NO_3$ is not lost via reaction with e.g. NO or if $N_2O_5$ loss rates are competitive. Note that use of $NO_3$ concentrations to calculate AN production rates would necessarily involve the assumption that all BVOC were measured and that the reaction rate constants and AN yields are accurately known.

Page 9, Line 24 and Line 29: Which days were included when calculating mean night-time mixing ratios? All days, only days including in Fig. 6, or some other combination?
This has been calculated for both scenarios and there is no significant difference in the average mixing ratios of the BVOCs is the humid nights are considered or not. However, to be rigorous, we have modified the text to clarify this and added a table with the rate coefficients and AN-yields for reaction of each individual BVOC measured with $NO_3$: The text has been modified as outlined above in the reply to general comment (1).

Page 9, Line 25: A citation for these alkyl nitrate yields should be given.
We had in fact already cited Atkinson and Arey (2003) and Perring et al (2013) (page 9, lines 19 and 20). However, we now use the IUPAC recommendations instead and quote associated uncertainties and spread in yields as assessed by IUPAC.

Page 9, Line 27-28: Some justification for assuming that heterogeneous N2O5 loss is minimal should be included.
We add the following justification: "This analysis implicitly assumes that indirect loss of $NO_3$ via the heterogeneous loss of $N_2O_5$ are insignificant compared to direct loss via reaction with BVOCs. This is expected for a forested region in summer and has been shown to be the case for the Taunus Observatorium (Crowley et al., 2010; Sobanski et al., 2016b) where, except for few occasions when the residual layer is sampled, the $NO_3$ lifetime with respect to gas-phase reactions with BVOCs is too short for indirect, heterogeneous loss to compete."

Page 9, Equation 2: See general comment 1.
See reply to general comment (1)

Page 10, Line 5: What uncertainty in the calculated value of OH does this correlation introduce?
Bonn et al report an uncertainty of a factor of two in the OH concentration. We have added this information to the text and also use it to calculate error bounds for the AN production

rate. We now write: "Bonn et al. (2014) report a maximum uncertainty of a factor two for [OH] derived in this manner. For an approximate estimate of day-time ANs prodution, we take the campaign mean mixing ratios of each VOC between 11:00 and 13:00 UTC as listed in Table 1. Based on the individual rate coefficients, alkyl nitrate yields and mean, noon-time concentrations of the VOCs measured (also listed in Table 1) and using equation (3);

$$P_{\Sigma ANs} = \Sigma_i \alpha_i k_{OH+RH_i}[OH][RH_i] \qquad (3)$$

we obtain a noon-time ANs production rate of $P(ANS)_{day} \approx 70^{+70}_{-35}$ pptv hr$^{-1}$ where the reported uncertainty is due only to the uncertainty in OH concentrations. "

Page 10, Line 6-7: Is the campaign mean calculated for all days or for only times included in Fig. 6?
This deals with the day versus nightime generation. The campaign means (e.g. of VOCs etc.) is for the entire campaign and not just for those with "dry" We now write: "For an approximate estimate of day-time ANs prodution, we take the campaign mean mixing ratios of each VOC between 11:00 and 13:00 UTC as listed in Table 1."

Page 10, Line 10, Equation 3: See general comment 1.
See reply to general comment (1).

Page 10, Line 20: I typically think of deposition dropping to near zero at night, since turbulent mixing is low. The authors should discuss further the likelihood of enhanced nighttime deposition.
This is true. The emphasis should actually be on the different chemical composition of the NO$_3$ generated night-time ANs compared to the OH generated daytime ones, rather than on the influence of turbulent transport. The day versus nightime production has been re-written as follows: "Although we calculate similar production rates of $\Sigma$ANs during the night when NO$_3$ is present, the daytime maximum in the $\Sigma$ANs mixing ratio is significantly larger, which has a number of likely causes. The first is related to missing OH reactivity, which, depending on the hydrocarbons involved, could potentially increase the OH-initiated rate of formation of $\Sigma$ANs yield by large factors. For example, if the hydrocarbons we measured would account for only 50 % of the OH reactivity and the missing ones were biogenic in nature (i.e. terpenoids with large AN-yields) we could expect more than a factor of two increase in calculated $P(\Sigma ANs)_{day}$. A further potential cause for larger daytime $\Sigma$ANs mixing ratios is a reduced loss of daytime $\Sigma$ANs with respect to chemical and depositional loss and condensation. This being a consequence of the different chemical composition and volatility of the ANs generated from NO$_3$- compared to OH-initiated oxidation. It is well established in chamber studies that the NO$_3$ induced oxidation of biogenics leads to highly functionalized ANs that partition largely to the aerosol phase and that the NO$_3$ oxidation of biogenic VOCs can lead to appreciable organo-nitrate content in atmospheric particulate matter (Fry et al., 2011; Fry et al., 2014; Boyd et al., 2015). Ambient measurements of aerosol composition show that night-time-generated organic nitrates formed in NO$_3$ + BVOC reactions are efficiently transferred to the condened phase (Rollins et al., 2012; Fry et al., 2013; Xu et al., 2015; Kiendler-Scharr et al., 2016), which is confirmed by modelled vapour pressures of the OH- and NO$_3$- initiated organic nitrate products from BVOC oxidation, the latter being substantially lower (Fry et al., 2013).

Table 3 shows that (of the BVOCs measured) limonene accounts for 40 % of the $NO_3$ loss rate, myrcene 30% and $\alpha$-pinene 29 %. Studies of the reaction between $\alpha$-pinene and $NO_3$ show that the yield of secondary organic aerosol (SOA) is less than 10 % (Hallquist et al., 1999; Perraud et al., 2010; Fry et al., 2014; Nah et al., 2016) with reports of the alkyl nitrates formed being exclusively in the gas-phase (Fry et al., 2014). This contrasts strongly with the situation for limonene, where SOA yields of up to $\approx$ 60 % have been reported, with more than 80 % of the alkyl nitrates formed being in the aerosol phase. There is no experimental data on SOA yields in the reaction between $NO_3$ and mycene nor on the gas-aerosol partitioning of the alkyl-nitrates formed. Recent experiments on $\beta$-pinene (Boyd et al., 2015) have shown that the SOA yield is neither strongly dependent on the relative concentrations of HO$x$ and NO$x$, which determines the nature of the end-products formed, nor on the relative humidity or seed-aerosol used. If we make the broad assumptions that 1) the SOA yield from mycene is the same as limonene and 2) that the fraction of ANs in the condensed phase is comparable to those found for limonene, we calculate that > 60 % of the alkyl nitrates formed at night at this site will be present in the aerosol phase and thus not detected by our instrument. In other words we would expect equation (3) to yield production rates of $\Sigma$ANs that exceed those derived from gas-phase $\Sigma$AN measurements by a factor between two and three.

We conclude that the apparent lower lifetime of night-time generated $\Sigma$ANs is thus likely to be the result of an increased fraction of low-volatility ANs gormed from terpenes initially reacting with $NO_3$ compared to OH-initiated oxidation, leading to a larger relative rate of SOA formation and partitioning of ANs to the condensed phase. (Fry et al., 2013) have shown that, at an urban / forested site in Colorado, the peak in particle phase organic nitrates occurs at night-time. The condensed phase ANs can undergo hydrolysis to $HNO_3$ (over a period few hours (Lee et al., 2016b)), and thus irreversible loss from the gas-phase, the latter enhanced by the lower temperatures and higher relative humidities encountered at night-time (Hallquist et al., 2009; Lee et al., 2016a)."

Page 10, Line 23-24: Can the SOA yields reported by Fry et al. 2011, 2014 be used to estimate the fraction of ANs produced that are likely to remain in the gas phase, and can that fraction be used to adjust Eq. 2 to describe only the gas-phase production of alkyl nitrates?

SOA yields of up to 60 % have been measured for limonene, which accounts for 40 % of the overall $NO_3$ reactivity (see new Table 3) but not for myrcene (30 % of the $NO_3$ reactivity). For $\beta$-pinene we know that the SOA yield is not strongly dependent on the relative humidity the seed aerosol or the relative HO$x$ to NO$x$ ratio (Boyd et al., 2015). We now add a rough estimate of the fraction of the alkyl nitrates that will be in the aerosol phase. The text now reads: "Table (3) shows that (of the BVOCs measured) limonene accounts for 40 % of the $NO_3$ loss rate, myrcene 30% and $\alpha$-pinene 29 %. Studies of the reaction between $\alpha$-pinene and $NO_3$ show that the yield of secondary organic aerosol (SOA) is less than 10 % (Hallquist et al., 1999; Perraud et al., 2010; Fry et al., 2014; Nah et al., 2016) with reports of the alkyl nitrates formed being exclusively in the gas-phase (Fry et al., 2014). This contrasts strongly with the situation for limonene, where SOA yields of up to $\approx$ 60 % have been reported, with more than 80 % of the alkyl nitrates formed being in the aerosol phase. There is no experimental data on SOA yields in the reaction between $NO_3$ and mycene or on the gas-aerosol partitioning of the alkyl-nitrtes formed. Recent experiments on $\beta$-pinene (Boyd et al., 2015) have shown that the SOA yield is neither strongly dependent on the relative concentrations of HO$x$ and NO$x$, which determines the nature of the end-products formed, nor on the

relative humidity or seed-aerosol used. If we make the broad assumptions that 1) the SOA yield from mycene is the same as limonene and 2) that the fraction of ANs in the condensed phase is comparable to those found for limonene, we calculate that > 60 % of the alkyl nitrates formed at night at this site will be present in the aerosol phase and thus not detected by our instrument. In other words we would expect equation (3) to yield production rates of ANs that exceed those derived from gas-phase AN measurements by a factor between two and three."

Page 11, Equation 5: The concentration of ozone and ANs should include the effect of chemical loss.
The equation has been modified to combine chemical and depositional losses:

$$\frac{\Delta O_3}{\Delta \Sigma ANs} = \frac{\int (P_{O3} - L_{O3} + E_{O3})dt}{\int (P_{\Sigma ANs} - L_{\Sigma ANs} + E_{\Sigma ANs})dt} \qquad \text{Eq. (5)}$$

In which $L$ represents loss terms (chemical and deposition) and $E$ represents entrainment, respectively.

Page 12, Line 3-7: Under the conditions of the NOTOMO campaign, what uncertainty in AN concentration does the correction procedure introduce?
The correction factor (0.8 to 1.5 with an average value of 1.1) and the total uncertainty in ANs for the NOTOMO campaign were discussed in Sobanski et al., 2016a. As the correction factor is $NO_x$ dependent and also $NO / NO_2$ ratio dependent it is not possible to quote a single uncertainty for the measurements. The following text has been added: "For NOTOMO, in which low mixing ratios of ANs were encountered, the vertical grouping of the data apparent in Fig. 7 (i.e. low resolution in concentration) is a result of the corrective procedure for extracting mixing ratios from raw data obtained in the hot and cold inlets, which involves iterative numerical simulation which converges when 1% agreement between observation and simulation is achieved. The average correction factor for ANs during NOTOMO was 1.1 (Sobanski et al., 2016a) and, based on a series of laboratory experiments, the ($NO_x$ dependent) uncertainty associated with this factor is expected to be less than 50 %."

Page 12, Line 20-22: This is likely an overestimate of the range of isoprene alkyl nitrate yield. Recent work on the isoprene branching ratio has generally found branching ratios on the higher end of this range (9-15%) (Teng et al., 2015, Xiong et al., 2015).
From their laboratory experiments, Xiong et al. report a value of $9^{+4}_{-3}$ %, in the centre of the range of 4-15 % reported in the literature. They also report 9-12 % from field data analysis. We have updated the text and now write: "Regarding the individual nitrate yields, some values used to calculate the average (see Table 1) are not precisely determined in the literature while others are estimated. For example, the most recent measurements of the yield of alkyl nitrates formed in the reaction of OH with isoprene in the presence of $NO_x$ (one of the best studied reaction systems) ranges from 6 to 13% (Xiong et al., 2015) and remains a significant source of uncertainty (Perring et al., 2013)."

Page 12, Line 33: Photolysis and chemical loss of alkyl nitrates is often a more important loss process than deposition (Xiong et al., 2015).
This can be true for isoprene (the subject of Xiong et al., 2015), especially under sunny, dry conditions. For nitrates derived from terpenes under conditions of low insolation and higher humidity (as found e.g. over long periods at the Kleiner Feldberg), the condensation /

deposition term is likely to gain in importance. We now write: "(Xiong et al., 2015) have shown that losses of isoprene derived nitrates (IN) due to reaction with OH (Lee et al., 2014) and photolysis can be rapid and start to deplete IN before photochemical production maximises at the peak of the daytime OH⁻ profile. They also indicate that early morning entrainment of IN from the residual layer can influence its diel profile. Along with photochemical degradation, dry deposition / hydrolysis can contribute to the alkyl nitrate sink, especially for those derived from biogenic VOCs (Jacobs et al., 2014; Rindelaub et al., 2015). Xiong et al. (2015), report efficient wet and dry deposition of nitrates derived from isoprene with significantly lower mixing ratios measured in conditions of reduced photochemical reactivity / rain."

Page 13, Line 6-9: Any explanation for the low concentrations of alkyl nitrates should also be able to explain the high concentration of ozone encountered during the NOTOMO campaign. Different air mass ages would potentially explain this as would different hydrocarbon mixtures. We have extended the text and added a new section on the differences between the PARADE and NOTOMO campaigns as outlined above in the response to general comment (2).

Page 13 Line 7: Given that on average, NOTOMO was warmer and sunnier than PARADE (page 8), what magnitude of changes in VOC emissions is expected between the two campaigns? Is this change large enough to explain the low observed yield of alkyl nitrates?
It is not only the prevailing temperature and insolation that control emission strengths and nature of BVOCs but also the time of year, availability of water and the growing phase (for different vegetation types) and general growth conditions in the proceeding months. As the two campaigns took place at different seasons, it is not necessarily given that the BVOC emissions will be the identical. We have extended the text and added a new section on the differences between the PARADE and NOTOMO campaigns as outlined above in the response to general comment (2).

Page 13 Line 8: Based on the mixture of non-biogenic VOCs measured during PARADE, would decreased concentrations of BVOCs lead to a lower average value of alpha? Is this value low enough to explain the observed O3-AN slope during NOTOMO?
First, we note that the previously reported mixing ratios of pentene and butadiene are considered to be erroneous (too high) and have been removed from the Table and the calculation of alpha has been repeated. We have added the following text: "To emphasise the different efficiency of AN production from VOCs of biogenic origin (BVOC) and anthropogenic origin (AVOC), Table 1 is separated into biogenic and anthropogenic VOCs and from the separate summed P(ANs) and P(O$_3$) we calculated the effective value of $\alpha(\text{OH})_{av}^{\text{VOCi}}$ that would have been obtained considering each class of VOC individually. The values thus obtained are $\alpha(\text{OH})_{av}^{\text{BVOCi}} = 17.3$ % and $\alpha(\text{OH})_{av}^{\text{AVOCi}} = 1.4$ %. Note that, for this particular hydrocarbon mixture, the O$_3$ production rates are much larger for the AVOCs (~ 1000 compared to ~400 for the BVOCS)."

Page 18, Line 20, Table 1: Several of the values in this table disagree with those listed in Perring et al, 2013, the listed source for the yield. This includes i-pentane(0.35/0.07), isoprene (0.044/0.07) and i-butane(0.255/0.096). The values in Table 1 should either be updated or new references given.
The values have been updated and the Table has been reorganised to separate the biogenic and anthropogenic VOCs.

**Technical Corrections:**

Page 1 Line 12: "Futher" should be "Further" Corrected
Page 6, Line 15: Extra period after Fig. 2. Corrected
Page 11, Line 14 Equation 4: There are some mis-matched parentheses in this equation. corrected
Page 18, Table 1: The mean noon time mixing ratio unit appears to be pptv, and not ppbv as written. Corrected.

---

## Author Comment (AC2) · 7 Feb 2017

In the following, the referee's comments are reproduced (black) along with our replies (blue) and changes made to the text (red) in the revised manuscript.

**General statement:**

This paper presents two summer field campaigns (2011 and 2015) of organic nitrate data at an urban influenced mountain site, focusing on measurements of alkyl nitrate formation and implications for production of ozone, and measurements of peroxynitrates, both using a thermal dissociation – cavity ringdown instrument, which detects these species via thermal dissociation and then detection of NO2. Reference is made to an earlier instrument paper for the measurement methodology and data corrections to account for some known interferences, the focus in this paper is in interpreting the field observations, including comparing across two years with different meteorological conditions.

This paper presents a novel combined dataset that will be of interest to the atmospheric chemistry community. I do see a few opportunities to extend the analysis and to compare to additional available measurements, which I mention below, and recommend publication after these revisions.

We thank the referee for this informed and thorough review and overall positive assessment of our manuscript. The manuscript has been improved in line with the comments listed below.

**Major suggestions:**

1) There are a few additional studies that I would suggest citing to inform your analysis, and to enable comparisons with your data:

• Kiendler -Scharr et al (2016) have just published a series of measurements of aerosol-phase RONO2 around Europe using AMS, and they comment on the ubiquity of $NO_3$ sourced nitrates – this would be a good point of comparison. ("Organic nitrates from night-time chemistry are ubiquitous in the European submicron aerosol," Geophys Res Lett, 10.1002/2016GL069239, 2016.)

This very recent paper is now also cited in the context of loss of ANs to the particle-phase: "It is well established in chamber studies that the NO3 induced oxidation of biogenics leads to highly functionalized ANs that partition largely to the aerosol phase and that the NO3 oxidation of biogenic VOCs can lead to appreciable organo-nitrate content in atmospheric particulate matter (Fry et al., 2011; Fry et al., 2014; Boyd et al., 2015). Ambient measurements of aerosol composition show that nightime-generated organic nitrates formed in NO3 + BVOC reactions are efficiently transferred to the condensed phase (Rollins et al., 2012; Fry et al., 2013; Xu et al., 2015; Kiendler-Scharr et al., 2016), which is confirmed by modelled vapour pressures of the OH- and NO3- initiated organic nitrate products from BVOC oxidation, the latter being substantially lower (Fry et al., 2013)."

• Fry et al. (2013) measured ΣPNs and ΣANs at an urban-influenced Colorado site, also in summertime, including doing the same O3/ ANs slope analysis to assess nitrate branching and relevance to $O_3$ formation. Compare yield and VOC mix? ("Observations of gas-and aerosol-phase organic nitrates at BEACHON-RoMBAS 2011," Atmos. Chem. Phys., 13, 8585-8605, 2013.)

We have added text describing the results and conclusions of this paper in different places. "Likewise, neglecting terms for entrainment and loss of ANs and $O_3$ will introduce a variable bias into calculations of $\alpha(OH)_{av}^{\Sigma ANs}$ . The loss of multifunctional ANs from terpene oxidation will bias the analysis to low vales of $\alpha(NO_3)$ (Fry et al., 2013). In the absence of information regarding the condensation rate or efficiency of deposition of a mixture of multi-functional nitrates or $O_3$ to the topographically complex terrain at the Taunus Observatory a more detailed analysis is not warranted.. The analysis does however make comparison with similar analyses for $\Sigma$ANs measurements possible, and our derived values of $\alpha(OH)_{av}^{\Sigma ANs}$ are consistent with those summarised by Perring et al (Perring et al., 2013) obtained both by observation of $\Sigma$ANs (7.1 % > $\alpha(OH)_{av}^{\Sigma ANs}$ > 0.8 % ) and calculated from VOC measurements (10.6 % > $\alpha(OH)_{av}^{VOCi}$ > 0.1 %) in various rural and urban locations. A similar analysis by Fry et al. (2013) of $\Sigma$AN and $O_3$ mixing ratios obtained during summer at a forest site with urban influence resulted in a value of $\alpha(OH)_{av}^{\Sigma ANs}$ = 2.9 %, intermediate between the value presented here for the PARADE and NOTOMO campaigns."

"Ambient measurements of aerosol composition show that nightime-generated organic nitrates formed in $NO_3$ + BVOC reactions are efficiently transferred to the condened phase (Rollins et al., 2012; Fry et al., 2013; Xu et al., 2015; Kiendler-Scharr et al., 2016), which is confirmed by modelled vapour pressures of the OH- and $NO_3$- initiated organic nitrate products from BVOC oxidation, the latter being substantially lower (Fry et al., 2013)."

"(Fry et al., 2013) have shown that, at an urban / forested site in Colorado, the peak in particle phase organic nitrates occurs at night-time."

• The Ng et al. 2016 review paper that you cite discussed some observations and general trends about organonitrate losses, including hydrolysis that would be valuable to consider in the context of your claims that nighttime nitrates may be lost more rapidly under damp and foggy conditions during NOTOMO, in contrast to the first year. There are several primary papers cited in Ng that might be relevant here, for example Boyd et al 2015 ("Secondary organic aerosol formation from the β-pinene+NO3 system: effect of humidity and peroxy radical fate," Atmos. Chem. Phys., 15, 7497-7522, 10.5194/acp-15-7497-2015, 2015.) –but in general I think the conclusions here would be at odds with your observations: nighttime $NO_3$ +terpene produced organonitrates are expected to be mostly primary and secondary nitrates, which appear to have longer lifetimes, not shorter, than daytime produced OH-initiated nitrates (which would include more tertiary nitrates).
To improve the discussion of the day-time versus night-time concentrations and losses of ANs we have adjusted the emphasis from hydrolysis (and its influence on deposition) to partitioning between the gas and aerosol phase. The section has been re-written: "Although we calculate similar production rates of $\Sigma$ANs during the night when $NO_3$ is present, the daytime maximum in the $\Sigma$ANs mixing ratio is significantly larger, which has a number of likely causes. The first is related to missing OH reactivity, which, depending on the hydrocarbons involved, could potentially increase the OH-initiated rate of formation of $\Sigma$ANs yield by large factors. For example, if the hydrocarbons we measured would account for only 50 % of the OH reactivity and the missing ones were biogenic in nature (i.e. terpenoids with large AN-yields) we could expect more than a factor of two increase in calculated $P(\Sigma ANs)_{day}$. A further potential cause for larger daytime $\Sigma$ANs mixing ratios is a reduced loss of daytime $\Sigma$ANs with respect to chemical and depositional loss and condensation. This being a consequence of the different chemical composition and volatility of the ANs generated from

NO$_3$- compared to OH-initiated oxidation. It is well established in chamber studies that the NO$_3$ induced oxidation of biogenics leads to highly functionalized ANs that partition largely to the aerosol phase and that the NO$_3$ oxidation of biogenic VOCs can lead to appreciable organo-nitrate content in atmospheric particulate matter (Fry et al., 2011; Fry et al., 2014; Boyd et al., 2015). Ambient measurements of aerosol composition show that nightime-generated organic nitrates formed in NO$_3$ + BVOC reactions are efficiently transferred to the condened phase (Rollins et al., 2012; Fry et al., 2013; Xu et al., 2015; Kiendler-Scharr et al., 2016), which is confirmed by modelled vapour pressures of the OH- and NO$_3$- initiated organic nitrate products from BVOC oxidation, the latter being substantially lower (Fry et al., 2013).

Table 3 shows that (of the BVOCs measured) limonene accounts for 40 % of the NO$_3$ loss rate, myrcene 30% and $\alpha$-pinene 29 %. Studies of the reaction between $\alpha$-pinene and NO$_3$ show that the yield of secondary organic aerosol (SOA) is less than 10 % (Hallquist et al., 1999; Perraud et al., 2010; Fry et al., 2014; Nah et al., 2016) with reports of the alkyl nitrates formed being exclusively in the gas-phase (Fry et al., 2014). This contrasts strongly with the situation for limonene, where SOA yields of up to $\approx$ 60 % have been reported, with more than 80 % of the alkyl nitrates formed being in the aerosol phase. There is no experimental data on SOA yields in the reaction between NO$_3$ and mycene nor on the gas-aerosol partitioning of the alkyl-nitrates formed. Recent experiments on $\beta$-pinene (Boyd et al., 2015) have shown that the SOA yield is neither strongly dependent on the relative concentrations of HO$x$ and NO$x$, which determines the nature of the end-products formed, nor on the relative humidity or seed-aerosol used. If we make the broad assumptions that 1) the SOA yield from mycene is the same as limonene and 2) that the fraction of ANs in the condensed phase is comparable to those found for limonene, we calculate that > 60 % of the alkyl nitrates formed at night at this site will be present in the aerosol phase and thus not detected by our instrument. In other words we would expect equation (3) to yield production rates of $\Sigma$ANs that exceed those derived from gas-phase $\Sigma$AN measurements by a factor between two and three.

We conclude that the apparent lower lifetime of night-time generated $\Sigma$ANs is thus likely to be the result of an increased fraction of low-volatility ANs gormed from terpenes initially reacting with NO$_3$ compared to OH-initiated oxidation, leading to a larger relative rate of SOA formation and partitioning of ANs to the condensed phase. (Fry et al., 2013) have shown that, at an urban / forested site in Colorado, the peak in particle phase organic nitrates occurs at night-time. The condensed phase ANs can undergo hydrolysis to HNO$_3$ (over a period few hours (Lee et al., 2016b)), and thus irreversible loss from the gas-phase, the latter enhanced by the lower temperatures and higher relative humidities encountered at night-time (Hallquist et al., 2009; Lee et al., 2016a)."

2) In my view, the weakest point in the paper is the explanation of the difference in PAN/ AN ratio and apparent O$_3$ AN branching ratio across the two years. It would be great to find more evidence to support and interpret this difference.
We have extended the discussion on this. In response to the comments below and those of reviewer 1,we have added a new section and now write:
**4.5.1 Inter-annual / seasonal differences in $\alpha$(OH), PARADE versus NOTOMO**

[revised manuscript text omitted]

• even though you don't have VOC measurements in year 2 (bummer!), could you look at e.g. temperature / sunshine differences correlated with VOCs measured within the PARADE period where you DO have the GCs running, and then extrapolate to the conditions during the second year of measurements?
Yes, the lack of biogenic VOCs or alkenes during NOTOMO is unfortunate. There are no permanent GC or PTRMS measurements at the site and the research group which normally does these measurements was participating in a parallel MPI-campaign. Some measurements of alkanes were taken, but these do not contribute significantly to AN generation at this site. While BVOC emissions are known to be temperature / sunlight dependent, they also depend on the seasonal growth stage and local weather in the preceding months. We have expanded the discussion of the differences in the two campaigns as described above.

• Or even use any other GC data taken at that site, whenever, to be able to say something about the potential range of year-over-year variability?
The long-term measurements at the site are meteorological in nature (operated by the German weather service) and some trace gases and particle-measurements by the HNLUG. There are no long term records of VOCs at this site. The PARADE campaign was the first, large intensive campaign at the site in which VOCs were measured using multiple instruments. We have expanded the discussion of the differences in the two campaigns as described above.

• Can you find literature to point to on how oxidized VOCs like nitrates deposition depends on met conditions (your claim at the top of p. 13)?
We now suggest that different air mass ages, hydrocarbon mix (and associated ANs formation rates and different rates of gas-to-particle conversion of the ANs) that all may contribute to the difference between the two years. We have expanded the discussion of the differences in the two campaigns as described above.

• Does the NO: $NO_2$ ratio during the two years support the apparent differences in PAN vs. ANs formation rate?
The noon-time $NO_2$-toNO ratios in PARADE and NOTOMO were 4.0 and 3.8, respectively and cannot explain the different PNs to ANs ratios. This information is now added: "A number of factors influence the relative concentrations of PNs and ANs. In general, higher temperatures are the result of higher levels of insolation and are thus usually related to higher $O_3$ concentrations and rates of photochemical processing of VOCs. This should lead to higher concentrations of both PNs and ANs. Higher levels of insolation will lead to higher NO to $NO_2$ ratios (noon-time $NO_2$-to-NO ratios were 4.0 (PARADE) and 3.8 (NOTOMO) and, given sufficient NO, elevated temperaures will reduce the lifetimes of PNs. Altogether, higher temperatures and more insolation favour AN production over PN production. During the two campaigns. This is essentially the opposite to what we observe and we conclude that other factors, including the mechanism of organic nitrate production from oxidation of different VOC types and rates of loss of the organic nitrates play a major role in contolling the relative abundance of ANs and PNs at this site (see below)."

• Can you find any NOx emissions data or traffic counts or similar to suggest that the NOy mix arriving at the site might be different across the 2 years?

We now mention that NO was lower during NOTOMO, consistent with (but not proving) the air being more aged. In the absence of NOy measurements, we cannot prove this. We write: "As the lifetime with respect to chemical loss/ deposition of alkyl nitrates derived from biogenic VOCs is expected to be shorter than that of $O_3$, the sampling of progressively aged air masses will bias $\alpha(\text{OH})_{av}^{\Sigma ANs}$ to low values when calculated from $O_3$ / $\Sigma AN$ correlations. A low value of $\alpha(\text{OH})_{av}^{\Sigma ANs}$ during NOTOMO could conceivably be the result of sampling on average older air masses than during PARADE. The lower NO$x$ levels in NOTOMO would support this contention, though in the absence of NO$y$ measurements, is not conclusive."

3) Figure suggestions

• It would be valuable to see some of the VOC variability in addition to reporting the mean noontime values in table 1. Could you add the reactively most important VOC or two to Fig. 1, to enable readers to see whether periods of high ANs/PNs correlate with higher VOC? Also, suggest to add the diurnally averaged version to Fig. 4 as well. Are all daytime-peaking or some nighttime? Could target trying to ID the dominant NO$_3$ +BVOC source of organonitrates at night vs. daytime RO2+NO source, which will help you put the ideas about hydrolysis lifetime and it's structure dependence in context.

Figure 1 has been expanded to show the time-series of isoprene (emission controlled by temperature and light, daytime peak) and α-pinene (emission rates controlled by temperature) as representatives of biogenic emissions. The diel profiles of isoprene and α-pinene are now also displayed in Figure 4.

• On Fig. 3, can you format the points so they don't obscure one another? it looks like the black points are behind the red, so it's hard to see their spread. Maybe use "+"s instead? Or bin /average data so there aren't so many points on the plot?

The solid circles have been replaced by + symbols.

• Please "squish" Fig. 4 and 5 on the horizontal axis (or equivalently, make them taller) so they are the same width as Fig. 6, where the diurnal pattern is easier to see because of the larger height to width aspect ratio.

Figs 4 and 5 have been stretched vertically.

• Suggest to rethink color scheme on figures. Red/black don't always means the same thing, leading to confusion. For example, could do dots vs solid for years, consistently, and always use color to refer to left/right axis?

Black and red are now used only to define different molecules / axes. The different campaign datasets in Fig 4 are now distinguished by line-type (solid or dashed).

• Suggest to add NO to figure 4

The diel average profile for NO has been added

• In caption to Fig. 5, briefly described how you separate out the rush-hour influenced Days.

This was done by close inspection of individual days and is mentioned in the text where this figure is discussed.

• Fig. 7: how did you choose 11-13 UTC for the $O_3$ vs ANs slopes? Did you check consistency using different time periods?

The period between 11-13 UTC was chosen is this corresponds to the maximum value of $J(O(^1D))$ and thus OH concentration at the site (see Fig. 4). In the text describing this figure we now write: For both the PARADE and NOTOMO campaigns, we analysed the mixing ratios of ANs and $O_3$ between 11:00 and 13:00 UTC (around the peak in $J(O(^1D))$ and thus OH levels) to calculate $\alpha(OH)_{av}^{\Sigma ANs}$.

Also, could the iterative correction procedure that makes the ANs data look binned on fig. 7 be the reason for lower ANs concentration measurements, too? What is the relative error on these measurements in each campaign, based on the correction procedure? Could you put error bars on these plots? (Again, might be best to bin first to avoid having a too-busy plot). The correction procedures and the related uncertainty are described in detail in the Thieser et al and Sobanski et al. The average correction factor was between 1.1 (NOTOMO) and 1.2 (PARADE) but with excursions at high NO$x$ levels to a factor of 2. The uncertainty on the correction factor has been estimated to by < 30 % (Thieser et al., 2016). We have added text in section 2.1 to mention this and now state that the difference between the years is significant. "The average correction factor for the ANs was 1.2, with maxumim values of 2. The uncertainty associated with the correction procedure is estimated as ~ 30 % (Thieser et al., 2016)." And later in the manuscript: "The difference in $\alpha(OH)_{av}^{\Sigma ANs}$ between PARADE ($7.2 \pm 0.5$ %) and NOTOMO (< 2 %) calculated from the $O_3$ and $\Sigma ANs$ datasets this manner is significant and cannot be explained by the uncertainty in the measurements of $O_3$ or $\Sigma ANs$ (see section (3.1). There are a number of potential causes for this difference between the two campaigns…………"

**Minor or technical edits:**

1) p. 2 line 1: "during the night (R6) (see below) to produce peroxy radicals which subsequently produce stable organonitrates by any radical terminal reaction. Organic peroxy radicals are also formed in..." correction made

2) p. 2 lin 12 "ultimate to $O_3$ formation." correction made

3) p. 2 line 31-32: suggest to include chemical formulae for each PAN, PPN, MPAN, analogous to how you show $RC(O)O_2NO_2$ on line 5 of this page. Addition made

4) p 3 line 4: "first measurements of total $\Sigma PNs$ and ..." correction made

5) p 3 line 32: clarify that the long observed $NO_3$ lifetime here is presumed due to low VOC mixing ratio – correct? If so, could you note the mixing ratio compared to another time where you're not sampling the residual layer?
We have added the lifetime information for "normal conditions". "On some nights during PARADE, the instruments sampled air from a low lying residual layer which resulted in very high $NO_3$ steady-state lifetimes ($\approx$ 1h). Otherwise the $NO_3$ lifetimes were generally less than 10 mins (Sobanski et al., 2016b)."

6) p 5 line 32 what does "HLUG" mean?
We have defined this in changed text: "Temperature, ozone, wind speed and wind direction data during NOTOMO were measured by the permanent instrumentation of the Hessian Agency for Nature Conservation, Environment and Geology (HLNUG) at this site."

7) p. 6 line 10-11. This is unclear. Add symbols to help reader correlate to table 1, e.g. "alky nitrate yields (alpha(ANs) and production rate of alkyl nitrates (P(ANs)) and ozone (P(O3)). The phrase "for ANs and PNs, respectively" I don't understand – P(O3) is just based on ANs – what do you mean by the PNs part?.

We have changed to text to clarify that the tabulated production rates are calculated from the OH concentrations and the VOC mixture: The VOCs measured during PARADE are listed in Table 1 along with their rate constants for reaction with OH $k_{OH}$) and also the associated alkyl nitrate yields, $\alpha(AN)$. We also list calculated production rates of ANs ($P(ANs)$), and $O_3$ ($P(O_3)$) derived from midday OH levels during PARADE.

8) p. 6 line 16. HLUG again - ?? and typo in "summarizes" correction made

9) suggest to find/replace "see later" and change to "see below", or vice versa, to make all consistent. We now use "see below" consistently.

10) p. 7 line 6: Maybe check if Lee et al's SOAS paper also does the sumANS / individual ANs comparison that Beaver's paper did? I think in the supplemental: http://www.pnas.org/content/113/6/1516
We now also cite Lee et al. when referring to CIMS measurements of speciated ANs.

11) p. 7 line 16-18: is this a chemical clock? Can you correlate with wind speed?
Given the complex terrain and heterogeneity of NO$x$ sources at the site, wind speed (or direction) will not correlate with chemical aging.

12) p. 7 line 22: "this site. Furthermore, the temperature dependence suggest that .." (because it's the T-dependent that suggests this, right? Not the f(NOx) values). Also, the last sentence of this paragraph offers an alternative explanation, so maybe start with "However, note that low temperatures also increase ..." and then put some more evidence for one of the other potential explanations, from Major Suggestions #1 above, below in this section.
Correction made as suggested. We write: "The high values of $f(NOx)$ during periods of low NO$x$ is the result of efficient conversion of NO$x$ to longer lived organic nitrates in photochemically aged air masses at this site. Furthermore, the temperature dependence indicates that organic nitrate formation is not limited by NO$x$. For a given level of NO$x$, $f(NOx)$ is larger when temperatures are higher, reflecting stronger biogenic emissions, and more intense photochemical activity and thus conversion rates of NO$x$ to organic nitrates (Olszyna et al., 1994; Day et al., 2008). However, note that low temperatures also increase the rate of transfer of soluble organic nitrates to the aerosol phase, which acts on $f(NOx)$ in the same direction."

13) p. 7 line 31: typo "emissions" corrected

14) p. 8 lin 1: unclear. do you mean the mean and daily maxima of each variable? reword.
We now write: "The campaign averaged, daily maxima in global radiation…."

15) p. 9 line 4: give number for the PARADE campaign too to compare.
Done. We now write: The most notable changes compared to Fig. 4 are the increase in [ΣANs] during the night. For the NOTOMO campaign, the night-time [ΣANs] represent ~60 % of the day-time value, for PARADE this is ~35%.

16) p. 9 line 15: typo "[NO3] and" corrected

17) p. 9 line 25-26: are you accounting for differences in rate here too, or assuming there is abundant $NO_3$ to fully oxidize all VOCs? Elaborate.

This text section has been extended to describe the calculation more fully, and a Table has been added. We now write: "Of the measured VOCs, isoprene, α-pinene, myrcene and limonene account for > 95% of the $NO_3$ reactivity and have substantial yields of ANs. The mean night-time mixing ratios for these four compounds during PARADE (excluding data where RH > 92 %) are listed in Table 3 along with the corresponding $NO_3$-reaction rate coefficients ($k$) and the alkyl nitrate yields (α) as reported by the IUPAC panel (IUPAC, 2016). The effective alkyl nitrate yield for this VOC mixture can be calculated from the relative flux of $NO_3$ reacting with each BVOC (depending on the BVOC mixing ratio and rate coefficient) and the alkyl nitrate yield for each individual BVOC. Note that, in the absence of laboratory investigations, the alkyl nitrate yield from $NO_3$ + myrcene is simply estimated as $50 \pm 30$ %, in line with other terpenes (IUPAC, 2016). As can be seen from Table 3, reactions with the measured terpenes dominate and the final, averaged yield of alkyl-nitrate is $\alpha(NO_3) = (0.41 \pm 0.31)$."

18) p. 10 line 15 typo "environment" corrected

19) p. 10 line 23: SOA yields could be even higher–see summary table in Ng 2016 paper.

This text section has been changed. We now write: "This contrasts strongly with the situation for limonene, where SOA yields of up to ≈ 60 % have been reported, with more than 80 % of the alkyl nitrates formed being in the aerosol phase. There is no experimental data on SOA yields in the reaction between $NO_3$ and mycene nor on the gas-aerosol partitioning of the alkyl-nitrates formed. Recent experiments on β-pinene (Boyd et al., 2015) have shown that the SOA yield is neither strongly dependent on the relative concentrations of HO$x$ and NO$x$, which determines the nature of the end-products formed, nor on the relative humidity or seed-aerosol used. If we make the broad assumptions that 1) the SOA yield from mycene is the same as limonene and 2) that the fraction of ANs in the condensed phase is comparable to those found for limonene, we calculate that > 60 % of the alkyl nitrates formed at night at this site will be present in the aerosol phase and thus not detected by our instrument."

20) p. 11 line 24: "than the loss terms, D and E, and Eq. 5 …"

The equation has been modified ($D$ is now $L$). We now write "In which $L$ represents loss terms (chemical and deposition) and $E$ represents entrainment, respectively. The ratio of $O_3$ to ΣANs after the OH oxidation of a VOC mixture has proceeded for a certain time, $dt$, is given by Eq. (5). At sufficiently high levels of OH, VOCs and NO, the photochemical production terms can be assumed to be larger than the loss or entrainment terms…"